# VISUAL REASONING BY PROGRESSIVE MODULE NETWORKS

**Seung Wook Kim**[1,2] , **Makarand Tapaswi**[1,2] , **Sanja Fidler**[1,2,3]
[1]Department of Computer Science, University of Toronto
[2]Vector Institute, Canada
[3]NVIDIA
`{seung,makarand,fidler}@cs.toronto.edu`

## ABSTRACT

Humans learn to solve tasks of increasing complexity by building on top of previously acquired knowledge. Typically, there exists a natural progression in the tasks that we learn – most do not require completely independent solutions, but can be broken down into simpler subtasks. We propose to represent a solver for each task as a neural module that calls existing modules (solvers for simpler tasks) in a functional program-like manner. Lower modules are a black box to the calling module, and communicate only via a query and an output. Thus, a module for a new task learns to query existing modules and composes their outputs in order to produce its own output. Our model effectively combines previous skill-sets, does not suffer from forgetting, and is fully differentiable. We test our model in learning a set of visual reasoning tasks, and demonstrate improved performances in all tasks by learning progressively. By evaluating the reasoning process using human judges, we show that our model is more interpretable than an attention-based baseline.

## 1 INTRODUCTION

Humans acquire skills and knowledge in a curriculum by building on top of previously acquired knowledge. For example, in school we first learn simple mathematical operations such as addition and multiplication before moving on to solving equations. Similarly, the ability to answer complex visual questions often requires the skills to understand attributes such as color, recognize a variety of objects, and be able to spatially relate them. Just like humans, machines may also benefit by sequentially learning tasks in progressive complexity and composing knowledge along the way.

The process of training a machine learning model to be able to solve multiple tasks, or *multi-task learning* (MTL), has been widely studied (Long et al., 2017; Ruder, 2017; Ruder et al., 2017; Rusu et al., 2016). The dominant approach is to have a model that shares parameters (*e.g.*, bottom layers of a CNN) with individualized prediction heads (Caruana, 1993; Long et al., 2017). By sharing parameters, models are able to learn better task-agnostic data representations. However, the tasks are disconnected as their outputs are not combined to solve tasks of increasing complexity. It is desirable if one task can learn to process the predictions from other tasks thereby reaping the benefits of MTL.

In this paper, we address the problem of MTL where tasks exhibit a natural progression in complexity. We propose Progressive Module Networks (PMN), a framework for multi-task learning by progressively designing modules on top of existing modules. Each module is a neural network that can query modules for lower-level tasks, which in turn may query modules for even simpler tasks. The modules communicate by learning to query other modules and process their outputs, while the internal module processes are a blackbox. This is similar to a computer program that uses available libraries without having to know their internal operations. Parent modules can choose which lower-level modules they want to query via a soft gating mechanism. Examining the queries, replies, and choices a parent module makes, we can understand the reasoning behind the module's output.

PMN is related but different from Andreas et al. (2016) and Hu et al. (2017). PMN's modules are task-level modules, and they are compositional, *i.e.* modules build on modules which build on modules. It allows efficient use of data by not needing to re-learn previously acquired knowledge. By learning selective information flow between modules, interpretability arises naturally.

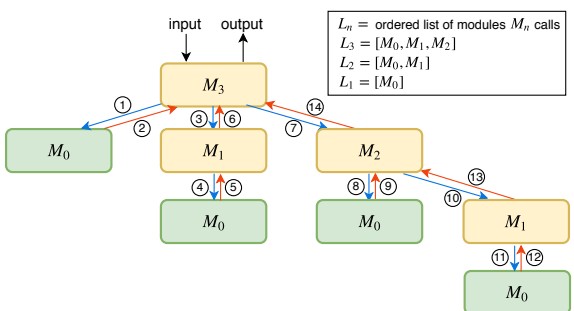

Figure 1: An example computation graph for PMN with four tasks. Green rectangles denote terminal modules, and yellow rectangles denote compositional modules. Blue arrows and red arrows represent calling and receiving outputs from submodules, respectively. White numbered circles denote computation order. For convenience, assume task levels correspond to the subscripts. Calling $M_3$ invokes a chain of calls (blue arrows) to lower level modules which stop at the terminal modules.

We demonstrate PMN in learning a set of visual reasoning tasks such as counting, captioning, and Visual Question Answering (VQA). PMN outperform baselines without module composition on all tasks. We further analyze the interpretability of PMN's reasoning process with human judges.

## 2 RELATED WORK

**Multi-task learning.** The dominant approach to multi-task learning is to have a model that shares parameters in a soft (Duong et al., 2015; Yang & Hospedales, 2017) or hard way (Caruana, 1993). Soft sharing refers to each task having independent weights that are constrained to be similar (*e.g.* via $\ell_2$ regularization (Duong et al., 2015), trace norm (Yang & Hospedales, 2017)) while hard sharing typically means that all tasks share the base network but have independent layers on the top (Kokkinos, 2017; Misra et al., 2016). While sharing parameters helps to compute a task-agnostic representation that is not overfit to a specific task, tasks do not directly share information or help each other.

Bilen & Vedaldi (2016) propose the Multinet architecture where tasks can interact with each other in addition to shared image features. Multinet solves one task at each time step and appends the encoded output of each task to existing data representation. A similar idea, Progressive Neural Networks (PNNs) (Rusu et al., 2016) use a new neural network for each task, but are designed to prevent catastrophic forgetting as they transfer knowledge from previous tasks by making lateral connections to representations of previously learned tasks. Recently, Wang et al. (2017) propose the VQA-Machine which exploits a set of existing algorithms to solve the VQA problem. Zamir et al. (2018) learn a computational taxonomy for task transfer learning on several vision problems. However, the major differences to this work are PMN's compositional modular structure, ability to directly query other modules, and the overall process of learning increasingly complex tasks.

**Module networks.** Pioneering work in modular structure, NMN (Andreas et al., 2016; Hu et al., 2017) addresses VQA where questions have a compositional structure. Given an inventory of small networks, or modules, NMN produces a layout for assembling the modules for any question. PMN is different from NMN as it can be easily extended to new tasks using its compositional structure. It also treats *each task* as modules which opens up promising ways to train models that can compartmentalize and perform multiple tasks as they progressively improve their abilities.

**Visual question answering.** VQA has seen great progress in recent years: improved multimodal pooling functions (Fukui et al., 2016; Kim et al., 2018), multi-hop attention (Yang et al., 2016), driving attention through both bottom-up and top-down schemes (Anderson et al., 2018), and modeling attention between words and image regions recurrently (Hudson & Manning, 2018) are some of the important advances. There are also attempts to generate programs or sequence of modules automatically that yield a list of interpretable steps (Hu et al., 2017; Johnson et al., 2017b) using policy gradient optimization. Our approach treats visual reasoning as a compositional multi-task problem, and shows that using sub-tasks compositionally can help improve performance and interpretability.

## 3 PROGRESSIVE MODULE NETWORKS

Most complex reasoning tasks can be broken down into a series of sequential reasoning steps. We hypothesize that there exists a hierarchy with regards to complexity and order of execution: high level tasks (*e.g.* counting) are more complex and benefit from leveraging outputs from lower level tasks (*e.g.* classification). For any task, Progressive Module Networks (PMN) learn a module that requests and uses outputs from lower modules to aid in solving the given task. **This process is compositional, *i.e.*, lower-level modules may call modules at an even lower level.** Solving a task corresponds to executing a directed acyclic computation graph where each node represents a module (see Fig. 1).

PMN has a plug-and-play architecture where modules can be replaced by their improved versions. This opens up promising ways to train intelligent robots that can compartmentalize and perform multiple tasks while progressively learning from experience and improving abilities. PMN also

chooses which lower level modules to use through a soft-gating mechanism. **A natural consequence of PMN's modularity and gating mechanism is interpretability.** While we do not need to know the internal workings of modules, we can examine the queries and replies along with the information about which modules were used to reason about why the parent module produced a certain output.

Formally, given a task $n$ at level $i$, the task module $M_n$ can query other modules $M_k$ at level $j$ such that $j < i$. Each module is designed to solve a particular task (output its best prediction) given an input and environment $\mathcal{E}$. Note that $\mathcal{E}$ is accessible to every module and represents a broader set of "sensory" information available to the model. For example, $\mathcal{E}$ may contain visual information such as an image, and text in the form of words (*i.e.*, question). PMN has two types of modules: (i) *terminal modules* execute the simplest tasks that do not require information from other modules (Sec. 3.1); and (ii) *compositional modules* that learn to efficiently communicate and exploit lower-level modules to solve a task (Sec. 3.2). We describe the tasks studied in this paper in Sec. 3.3 and provide a detailed example of how PMN is implemented and executed for VQA (Sec. 3.4).

## 3.1 TERMINAL MODULES

Terminal modules are by definition at the lowest level $0$. They are analogous to base cases in a recursive function. Given an input query $q$, a terminal module $M_\ell$ generates an output $o = M_\ell(q)$, where $M_\ell$ is implemented with a neural network. A typical example of a terminal module is an object classifier that takes as input a visual descriptor $q$, and predicts the object label $o$.

## 3.2 COMPOSITIONAL MODULES

A compositional module $M_n$ makes a sequence of calls to lower level modules which in turn make calls to their children, in a manner similar to depth-first search (see Fig. 1). We denote the list of modules that $M_n$ is allowed to call by $\mathcal{L}_n = [M_m, \ldots, M_l]$. Every module in $\mathcal{L}_n$ has level lower than $M_n$. Since lower modules need not be sufficient in fully solving the new task, we optionally include a terminal module $\Delta_n$ that performs "residual" reasoning. Also, many tasks require an attention mechanism to focus on certain parts of data. We denote $\Omega_n$ as a terminal module that performs such soft-attention. $\Delta_n$ and $\Omega_n$ are optionally inserted to the list $\mathcal{L}_n$ and treated as any other module.

The compositional aspect of PMN means that modules in $\mathcal{L}_n$ can have their own hierarchy of calls. We make $\mathcal{L}_n$ an ordered list, where calls are being made in a sequential order, starting with the first in the list. This way, information produced by earlier modules can be used when generating the query for the next. For example, if one module is performing object detection, we may want to use its output (bounding box proposals), for querying other modules such as an attribute classifier.

For this work, the list $\mathcal{L}_n$, and thus the levels of tasks, are determined by hand. Relaxing this and letting the model learn the task hierarchy itself is a challenging direction that we leave for future work. Also, notice that the number of back-and-forth communications increases exponentially if each module makes use of every lower-level module. Thus, in practice we restrict the list $\mathcal{L}_n$ to those lower-level modules that may intuitively be needed by the task. We emphasize that $M_n$ still (softly) chooses between them, and thus the expert intervention only removes the lower-level modules that are uninformative to the task.

Our compositional module $M_n$ runs (pre-determined) $T_n$ passes over the list $\mathcal{L}_n$. It keeps track of a state variable $s^t$ at time step $t \leq T_n$. This contains useful information obtained by querying other modules. For example, $s^t$ can be the hidden state of a Recurrent Neural Network. Each time step corresponds to executing *every* module in $\mathcal{L}_n$ and updating the state variable. We describe the module components below, and Algorithm 1 shows how the computation is performed. An example implementation of the components and demonstration of how they are used is detailed in Sec. 3.4.

**State initializer.** Given a query (input) $q_n$, the initial state $s^1$ is produced using a *state initializer* $I_n$.

**Importance function.** For each module $M_k$ (and $\Delta_n$, $\Omega_n$) in $\mathcal{L}_n$, we compute an importance score $g_n^k$ with $G_n(s^t)$. The purpose of $g_n^k$ is to enable $M_n$ to (softly) choose which modules to use. This also enables training all module components with backpropagation. Notice that $g_n^k$ is input dependent, and thus the module $M_n$ can effectively control which lower-level module outputs to use in state $s^t$. Here, $G_n$ can be implemented as an MLP followed by either a softmax over submodules, or a sigmoid that outputs a score for each submodule. However, note that the proposed setup can be modified to adopt hard-gating mechanism using a threshold or sampling with reinforcement learning.

---

**Algorithm 1** Computation performed by our Progressive Module Network, for one module $M_n$

---

1: **function** $M_n(q_n)$      ▷ The environment $\mathcal{E}$ and module list $\mathcal{L}_n$ are global variables
2:     $s^1 = I_n(q_n)$      ▷ initialize the state variable
3:     **for** $t \leftarrow 1$ to $T_n$ **do**      ▷ $T_n$ is the maximum time step
4:        $V = []$      ▷ wipe out scratch pad $V$
5:        $g_n^1, \ldots, g_n^{|\mathcal{L}_n|} = G_n(s^t)$      ▷ compute importance scores
6:        **for** $k \leftarrow 1$ to $|\mathcal{L}_n|$ **do**      ▷ $\mathcal{L}_n$ is the sequence of lower modules $[M_m, ..., M_l]$
7:           $q_k = Q_{n \rightarrow k}(s^t, V, G_n(s^t))$      ▷ produce query for $M_k$
8:           $o_k = \mathcal{L}_n[k](q_k)$      ▷ call $k^{th}$ module $M_k = \mathcal{L}_n[k]$, generate output
9:           $v_k = R_{k \rightarrow n}(s^t, o_k)$      ▷ receive and project output
10:          $V.\text{append}(v_k)$      ▷ write $v_k$ to pad $V$
11:        $s^{t+1} = U_n(s^t, V, \mathcal{E}, G_n(s^t))$      ▷ update module state
12:     $o_n = \Psi_n(s^1, \ldots, s^{T_n}, q_n, \mathcal{E})$      ▷ produce the output
13:     **return** $o_n$

---

**Query transmitter and receiver.** A query for module $M_k$ in $\mathcal{L}_n$ is produced using a *query transmitter*, as $q_k = Q_{n \rightarrow k}(s^t, V, G_n(s^t))$. The output $o_k = M_k(q_k)$ received from $M_k$ is modified using a *receiver function*, as $v_k = R_{k \rightarrow n}(s^t, o_k)$. One can think of these functions as translators of the inputs and outputs into the module's own "language". Note that each module has a scratch pad $V$ to store outputs it receives from a list of lower modules $\mathcal{L}_n$, *i.e.*, $v_k$ is stored to $V$. $Q_{n \rightarrow k}$ and $R_{k \rightarrow n}$ stand for the query transmitter from task $n$ to task $k$ and receiver from task $k$ to task $n$, respectively.

**State update function.** After every module in $\mathcal{L}_n$ is executed, module $M_n$ updates its internal state using a *state update function* $U_n$ as $s^{t+1} = U_n(s^t, V, \mathcal{E}, G_n(s^t))$. This completes one time step of the module's computation. Once the state is updated, the scratch pad $V$ is wiped clean and is ready for new outputs. An example can be a simple gated sum of all outputs, *i.e.*, $s^{t+1} = \sum_k g_n^k \cdot v_k$.

**Prediction function.** After $T_n$ steps, the final module output is produced using a *prediction function* $\Psi_n$ as $o_n = \Psi_n(s^1, \ldots, s^{T_n}, q_n, \mathcal{E})$. Recall that $\mathcal{E}$ is the environment.

All module functions: state initializer $I$, importance function $G$, query transmitter $Q$, receiver $R$, state update function $U$, residual module $\Delta$, attention module $\Omega$, and prediction function $\Psi$ are implemented as neural networks or simple assignment functions (*e.g.* set $q_k = v_l$). Note that all variables (*e.g.* $o_k, q_k, v_k, s^t$) are continuous vectors to allow learning with standard backpropagation. For example, the output of the relationship detection module that predicts an object bounding box is a $N$ dimensional soft-maxed vector (assuming there are total of $N$ boxes or image regions in $\mathcal{E}$).

**Training.** We train our modules sequentially, from low level to high level tasks, one at a time. The internal weights of the lower level modules are not updated, thus preserving their performance on the original task. The new module only learns to communicate with them via the query transmitter $Q$ and receiver $R$. We do train the weights of $\Delta$ and $\Omega$. We train $I$, $G$, $Q$, $R$, $U$, and $\Psi$, by allowing gradients to pass through the lower level modules. The loss function depends on the task $n$.

### 3.3 Progressive Module Networks for Visual Reasoning

We present an example of how PMN can be adopted for several tasks related to visual reasoning. In particular, we consider six tasks: object classification, attribute classification, relationship detection, object counting, image captioning, and visual question answering. Our environment $\mathcal{E}$ consists of: **(i)** *image regions*: $N$ image features $X = [X_1, \ldots, X_N]$, each $X_i \in \mathbb{R}^d$ with corresponding bounding box coordinates $\mathbf{b} = [b_1, \ldots, b_N]$ extracted from Faster R-CNN (Ren et al., 2015); and **(ii)** *language*: vector representation of a sentence $S$ (in our example, a question). $S$ is computed through a Gated Recurrent Unit (Cho et al., 2014) by feeding in word embeddings $[w_1, \ldots, w_T]$ at each time step.

Below, we discuss each task and a module designed to solve it. We provide detailed implementation and execution process of the VQA module in Sec. 3.4. For other modules, we present a brief overview of what each module does in this section. Further implementation details of all module architectures are in Appendix A.

**Object and Attribute Classification (level 0).** Object classification is concerned with naming the object that appears in the image region, while attribute classification predicts the object's attributes (*e.g.* color). As these two tasks are fairly simple (not necessarily easy), we place $M_{\text{obj}}$ and $M_{\text{att}}$ as

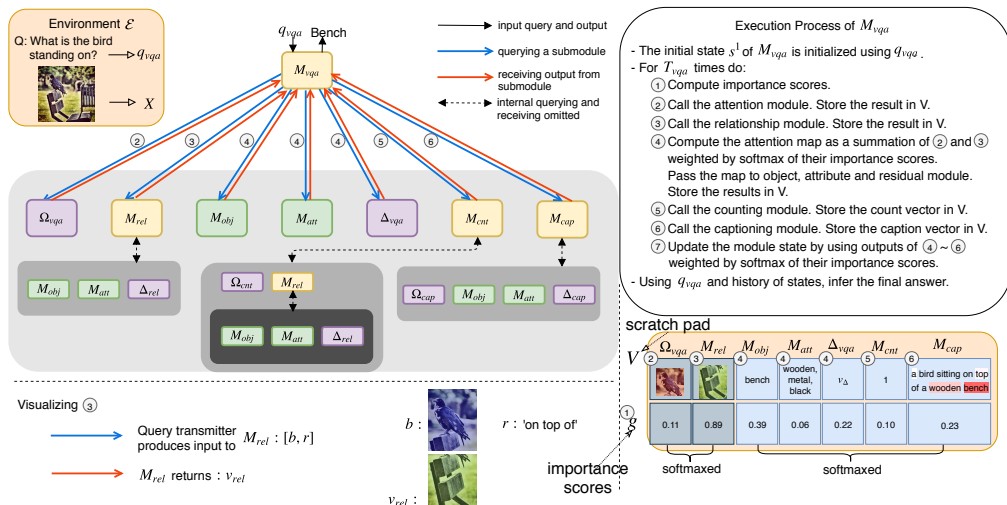

Figure 2: Example of PMN's module execution trace on the VQA task. Numbers in circles indicate the order of execution. Intensity of gray blocks represents depth of module calls. All variables including queries and outputs stored in $V$ are continuous vectors to allow learning with standard backpropagation (*e.g.*, caption is composed of a sequence of softmaxed $W$ dimensional vectors for vocabulary size $W$). For $M_{cap}$, words with higher intensity in red are deemed more relevant by $R_{vqa}^{cap}$. **Top:** high level view of module execution process. **Bottom right:** computed importance scores and populated scratch pad. Note that we perform the first softmax operation on $(\Omega_{vqa}, M_{rel})$ to obtain an attention map and the second on $(M_{obj}, M_{att}, \Delta_{vqa}, M_{cnt}, M_{cap})$ to obtain the answer. **Bottom left:** visualizing the query $M_{vqa}$ sends to $M_{rel}$, and the received output.

terminal modules at level 0. $M_{obj}$ consists of an MLP that takes as input a visual descriptor for a bounding box $b_i$, *i.e.*, $q_{obj} = X_i$, and produces $o_{obj} = M_{obj}(q_{obj})$, the penultimate vector prior to classification. Attribute module $M_{att}$ has a similar structure. These are the only modules for which we do not use actual output labels, as we obtained better results for higher level tasks empirically.

**Image Captioning (level 1).** In image captioning, one needs to produce a natural language description of the image. We design our module $M_{cap}$ as a compositional module that uses information from $\mathcal{L}_{cap} = [\Omega_{cap}, M_{obj}, M_{att}, \Delta_{cap}]$. We implement the state update function as a two-layer GRU network with $s^t$ corresponding to the hidden states. Similar to Anderson et al. (2018), at each time step, the attention module $\Omega_{cap}$ attends over image regions $X$ using the hidden state of the first layer. The attention map $m$ is added to the scratch pad $V$. The query transmitters produce a query (image vector at the attended location) using $m$ to obtain nouns $M_{obj}$ and adjectives $M_{att}$. The residual module $\Delta_{cap}$ processes other image-related semantic information. The outputs from modules in $\mathcal{L}_{cap}$ are projected to a common vector space (same dimensions) by the receivers and stored in the scratch pad. Based on their importance score, the gated sum of the outputs is used to update the state. The natural language sentence $o_{cap}$ is obtained by predicting a word at each time step using a fully connected layer on the hidden state of the second GRU layer.

**Relationship Detection (level 1).** In this task the model is expected to produce triplets in the form of "subject - relationship - object" (Lu et al., 2016). We re-purpose this task as one that involves finding the relevant item (region) in an image that is related to a given input through a given relationship. The input to the module is $q_{rel} = [b_i, r]$ where $b_i$ is a one-hot encoding of the input box and $r$ is a one-hot encoding of the relationship category (*e.g.* above, behind). The module produces $o_{rel} = b_{out}$ corresponding to the box for the subject/object related to the input $b_i$ through $r$. We place $M_{rel}$ on the first level as it may use object and attribute information that can be useful to infer relationships, *i.e.*, $\mathcal{L}_{rel} = [M_{obj}, M_{att}, \Delta_{rel}]$. We train the module using the cross-entropy loss.

**Object Counting (level 2).** Our next task is counting the number of objects in the image. Given a vector representation of a natural language question (*e.g.* how many cats are in this image?), the goal of this module is to produce a numerical count. The counting task is at a higher-level since it may also require us to understand relationships between objects. For example, "how many cats are on the blue chair?", requires counting cats *on top of* the blue chair. We thus place $M_{cnt}$ on the second level and provide it access to $\mathcal{L}_{cnt} = [\Omega_{cnt}, M_{rel}]$. The attention module $\Omega_{cnt}$ finds relevant objects by using the input question vector. $M_{cnt}$ may also query $M_{rel}$ if the question requires relational reasoning. To answer "how many cats are on the blue chair", we can expect the query transmitter

$Q_{\text{cnt}\to\text{rel}}$ to produce a query $q_{\text{rel}} = [b_i, r]$ for the relationship module $M_{\text{rel}}$ that includes the chair bounding box $b_i$ and relationship "on top of" $r$ so that $M_{\text{rel}}$ outputs boxes that contain cats on the chair. Note that both $\Omega_{\text{cnt}}$ and $M_{\text{rel}}$ produce attention maps on the boxes. The state update function softly chooses a useful attention map by calculating softmax on the importance scores of $\Omega_{\text{cnt}}$ and $M_{\text{rel}}$. For prediction function $\Psi_{\text{cnt}}$, we adopt the counting algorithm by Zhang et al. (2018), which builds a graph representation from attention maps to count objects. $M_{\text{cnt}}$ returns $o_{\text{cnt}}$ which is the count vector corresponding to softmaxed one-hot encoding of the count (with maximum count $\in \mathbb{Z}$).

**Visual Question Answering (level 3).** VQA is our final and most complex task. Given a vector representation of a natural language question, $q_{\text{vqa}}$, the VQA module $M_{\text{vqa}}$ uses $\mathcal{L}_{\text{vqa}} = [\Omega_{\text{vqa}}, M_{\text{rel}}, M_{\text{obj}}, M_{\text{att}}, \Delta_{\text{vqa}}, M_{\text{cnt}}, M_{\text{cap}}]$. Similar to $M_{\text{cnt}}$, $M_{\text{vqa}}$ makes use of $\Omega_{\text{vqa}}$ and $M_{\text{rel}}$ to get an attention map. The produced attention map is fed to the downstream modules $[M_{\text{obj}}, M_{\text{att}}, \Delta_{\text{vqa}}]$ using the query transmitters. $M_{\text{vqa}}$ also queries $M_{\text{cnt}}$ which produces a count vector. For the last entry $M_{\text{cap}}$ in $\mathcal{L}_{\text{vqa}}$, the receiver attends over the words of the entire caption produced by $M_{\text{cap}}$ to find relevant answers. The received outputs are used depending on the importance scores. Finally, $\Psi_{\text{vqa}}$ produces an output vector based on $q_{\text{vqa}}$ and all states $s^t$.

### 3.4 EXAMPLE: $M_{\text{vqa}}$ FOR VISUAL QUESTION ANSWERING

We give a detailed example of how PMN is implemented for the VQA task. The entire execution process is depicted in Fig. 2, and the general algorithm is tabulated in Alg. 1.

The input $q_{\text{vqa}}$ is a vector representing a natural language question (*i.e.* the sentence vector $S \in \mathcal{E}$). The state variable $s^t$ is represented by a tuple $(q_{\text{vqa}}^t, k^{t-1})$ where $q_{\text{vqa}}^t$ represents query to ask at time $t$ and $k^{t-1}$ represents knowledge gathered at time $t-1$. The state initializer $I_{\text{vqa}}$ is composed of a GRU with hidden state dimension 512. The first input to GRU is $q_{\text{vqa}}$, and $I_{\text{vqa}}$ sets $s^1 = (q_{\text{vqa}}^1, \mathbf{0})$ where $q_{\text{vqa}}^1$ is the first hidden state of the GRU and $\mathbf{0}$ is a zero vector (no knowledge at first).

For $t$ in $T_{\text{vqa}} = 2$, $M_{\text{vqa}}$ does the following seven operations:

(1) The importance function $G_{\text{vqa}}$ is executed. It is implemented as a linear layer $\mathbb{R}^{512} \to \mathbb{R}^7$ (for the seven modules in $\mathcal{L}_{\text{vqa}}$) that takes $s^t$, specifically $q_{\text{vqa}}^t \in s^t$ as input.

(2) $Q_{\text{vqa}\to\Omega}$ passes $q_{\text{vqa}}^t$ to the attention module $\Omega_{\text{vqa}}$ which attends over the image regions $X$ with $q_{\text{vqa}}^t$ as the key vector. $\Omega_{\text{vqa}}$ is implemented as an MLP that computes a dot-product soft-attention similar to Yang et al. (2016). The returned attention map $v_\Omega$ is added to the scratch pad $V$.

(3) $Q_{\text{vqa}\to\text{rel}}$ produces an input tuple $[b, r]$ for $M_{\text{rel}}$. The input object box $b$ is produced by a MLP that does soft attention on image boxes, and the relationship category $r$ is produced through a MLP with $q_{\text{vqa}}^t$ as input. $M_{\text{rel}}$ is called with $[b, r]$ and the returned map $v_{\text{rel}}$ is added to $V$.

(4) $Q_{\text{vqa}\to\text{obj}}$, $Q_{\text{vqa}\to\text{att}}$, and $Q_{\text{vqa}\to\Delta}$ first compute a joint attention map $m$ as summation of $(v_\Omega, v_{\text{rel}})$ weighted by the softmaxed importance scores of $(\Omega_{\text{vqa}}, M_{\text{rel}})$, and they pass the sum of visual features $X$ weighted by $m$ to the corresponding modules. $\Delta_{\text{vqa}}$ is implemented as an MLP. The receivers project the outputs into 512 dimensional vectors $v_{\text{obj}}, v_{\text{att}}$, and $v_\Delta$ through a sequence of linear layers, batch norm, and $\tanh()$ nonlinearities. They are added to $V$.

(5) $Q_{\text{vqa}\to\text{cnt}}$ passes $q_{\text{vqa}}^t$ to $M_{\text{cnt}}$ which returns $o_{\text{cnt}}$. $R_{\text{cnt}\to\text{vqa}}$ projects the count vector $o_{\text{cnt}}$ into a 512 dimensional vector $v_{cnt}$ through the same sequence of layers as above. $v_{\text{cnt}}$ is added to $V$.

(6) $M_{\text{vqa}}$ calls $M_{\text{cap}}$ and $R_{\text{cap}\to\text{vqa}}$ receives natural language caption of the image. It converts words in the caption into vectors $[w_1, \ldots, w_T]$ through an embedding layer. The embedding layer is initialized with 300 dimensional GloVe vectors (Pennington et al., 2014) and fine-tuned. It does softmax attention operation over $[w_1, \ldots, w_T]$ through a MLP with $q_{\text{vqa}}^t \in s_t$ as the key vector, resulting in word probabilities $p_1, \ldots, p_T$. The sentence representation $\sum_i^T p_i \cdot w_i$ is projected into a 512 dimensional vector $v_{\text{cap}}$ using the same sequence as $v_{\text{cnt}}$. $v_{\text{cap}}$ is added to $V$.

(7) The state update function $U_{\text{vqa}}$ first does softmax operation over the importance scores of $(M_{\text{obj}}, M_{\text{att}}, \Delta_{\text{vqa}}, M_{\text{cnt}}, M_{\text{cap}})$. We define an intermediate knowledge vector $k^t$ as the summation of $(v_{\text{obj}}, v_{\text{att}}, \delta_{\text{vqa}}, v_{\text{cnt}}, v_{\text{cap}})$ weighted by the softmaxed importance scores. $U_{\text{vqa}}$ passes $k^t$ as input to the GRU initialized by $I_{\text{vqa}}$, and we get $q_{\text{vqa}}^{t+1}$ the new hidden state of the GRU. The new state $s^{t+1}$ is set to $(q_{\text{vqa}}^{t+1}, k^t)$. This process allows the GRU to compute new question and state vectors based on what has been *asked* and *seen*.

After $T_{\text{vqa}}$ steps, the prediction function $\Psi_{\text{vqa}}$ computes the final output based on the initial question vector $q_{\text{vqa}}$ and all knowledge vectors $k^t \in s^t$. Here, $q_{\text{vqa}}$ and $k^t$ are fused with gated-tanh layers and fed through a final classification layer similar to Anderson et al. (2018), and the logits for all time steps are added. The resulting logit is the final output $o_{\text{vqa}}$ that corresponds to an answer in the vocabulary of the VQA task. Note that the exact form of each module can be different. While we leave a more general architecture across tasks as future work, we stress that one of PMN's strengths is that once a module is trained, it can be used as a blackbox by the higher-level modules. Details of other modules' architectures are provided in Appendix A.

## 4 EXPERIMENTS

We present experiments demonstrating the impact of progressive learning of modules. We also analyze and evaluate the reasoning process of PMN as it is naturally interpretable. We conduct experiments on three datasets (see Appendix B.1 for details): Visual Genome (VG) (Krishna et al., 2016), VQA 2.0 (Goyal et al., 2017), MS-COCO (Lin et al., 2014). These datasets contain natural images and are thus more complex in visual appearance and language diversity than CLEVR (Johnson et al., 2017a) that contains synthetic scenes. Neural module networks (Andreas et al., 2016; Hu et al., 2017) show excellent performance on CLEVR but their performance on natural images is quite below the state-of-the-art. For all datasets, we extract bounding boxes and their feature representations using a pretrained model from Anderson et al. (2018).

### 4.1 PROGRESSIVE LEARNING OF TASKS AND MODULES

**Object and Attribute Classification.** We train these modules with annotated bounding boxes from the VG dataset. We follow Anderson et al. (2018) and use 1,600 and 400 most commonly occurring object and attribute classes, respectively. Each extracted box is associated with the ground truth label of the object with greatest overlap. It is ignored if there are no ground truth boxes with IoU > 0.5. This way, each box is annotated with one object label and zero or more attribute labels. $M_{\text{obj}}$ achieves 54.9% top-1 accuracy and 86.1% top-5 accuracy. We report mean average precision (mAP) for attribute classification which is a multi-label classification problem. $M_{\text{att}}$ achieves 0.14 mAP and 0.49 weighted mAP. *mAP* is defined as the mean over all classes, and *weighted mAP* is weighted by the number of instances for each class. As there are a lot of redundant classes (*e.g.* car, cars, vehicle) and boxes have sparse attribute annotations, the accuracy may seem artificially low.

**Image Captioning.** We report results on MS-COCO for image captioning. We use the standard split from the 2014 captioning challenge to avoid data contamination with VQA 2.0 or VG. This split contains 30% less data compared to that proposed in Karpathy & Fei-Fei (2015) that most current works adopt. We report performance using the CIDEr (Vedantam et al., 2015) metric. A baseline (non-compositional module) achieves a strong CIDEr score of 108. Using the object and attribute modules we are able to obtain 109 CIDEr. While this is not a large improvement, we suspect a reason for this is the limited vocabulary. The MS-COCO dataset has a fixed set of 80 object categories and does not benefit by using knowledge from modules that are trained on more diverse data. We believe the benefits of PMN would be clearer on a diverse captioning dataset with many more object classes. Also, including high-level modules such as $M_{\text{vqa}}$ would be an interesting direction for future work.

**Relationship Detection.** We use top 20 commonly occurring relationship categories, which are defined by a set of words with similar meaning (*e.g.* in, inside, standing in). Relationship tuples in the form of "subject - relationship - object" are extracted from Visual Genome (Krishna et al., 2016; Lu et al., 2016). We train and validate the relationship detection module using 200K/38K train/val tuples that have both subject and object boxes overlapping with the ground truth boxes (IoU > 0.7). Table 1 shows improvement in performance when using modules. Even though accuracy is relatively low, model errors are reasonable, qualitatively. This is partially attributed to multiple correct answers although there is only one ground truth answer.

Table 1: Performance of $M_{\text{rel}}$

| Model | Composition | | | Acc. (%) | |
|---|---|---|---|---|---|
| | BASE | OBJ | ATT | Object | Subject |
| $M_{\text{rel}_0}$ | ✓ | - | - | 51.0 | 55.9 |
| $M_{\text{rel}_1}$ | ✓ | $M_{\text{obj}}$ | $M_{\text{att}}$ | 53.4 | 57.8 |

Table 2: Accuracy for $M_{\text{cnt}}$

| Model | Composition | | | | Acc. (%) |
|---|---|---|---|---|---|
| | BASE | OBJ | ATT | REL | |
| $M_{\text{cnt}_0}$ | ✓ | - | - | - | 45.4 |
| $M_{\text{cnt}_1}$ | ✓ | $M_{\text{obj}}$ | $M_{\text{att}}$ | - | 47.4 |
| $M_{\text{cnt}_2}$ | ✓ | $M_{\text{obj}}$ | $M_{\text{att}}$ | $M_{\text{rel}_1}$ | 50.0 |

**Object Counting.** We extract questions starting with 'how many' from VQA 2.0 which results in a training set of ∼50K questions. We additionally create ∼89K synthetic questions based on the VG dataset by counting the object boxes and forming 'how many'

Table 3: Model ablation for VQA. We report mean±std computed over three runs. Steady increase indicates that information from modules helps, and that PMN makes use of lower modules effectively. The base model $M_{\text{vqa}_0}$ does not use any lower level modules other than the residual and attention modules.

| Model | BASE | Composition | | | | | Accuracy (%) |
|---|---|---|---|---|---|---|---|
| | | OBJ | ATT | REL | CNT | CAP | |
| $M_{\text{vqa}_0}$ | ✓ | - | - | - | - | - | 62.05 ±0.11 |
| $M_{\text{vqa}_1}$ | ✓ | $M_{\text{obj}}$ | $M_{\text{att}}$ | - | - | - | 63.38 ±0.05 |
| $M_{\text{vqa}_2}$ | ✓ | $M_{\text{obj}}$ | $M_{\text{att}}$ | $M_{\text{rel}_1}$ | - | - | 63.64 ±0.07 |
| $M_{\text{vqa}_3}$ | ✓ | $M_{\text{obj}}$ | $M_{\text{att}}$ | - | $M_{\text{cnt}_1}$ | - | 64.06 ±0.05 |
| $M_{\text{vqa}_4}$ | ✓ | $M_{\text{obj}}$ | $M_{\text{att}}$ | $M_{\text{rel}_1}$ | $M_{\text{cnt}_2}$ | - | 64.36 ±0.06 |
| $M_{\text{vqa}_5}$ | ✓ | $M_{\text{obj}}$ | $M_{\text{att}}$ | $M_{\text{rel}_1}$ | $M_{\text{cnt}_2}$ | $M_{\text{cap}_1}$ | 64.68 ±0.04 |

Table 4: Comparing VQA accuracy of PMN with state-of-the-art models. Rows with Ens ✓denote ensemble models. test-dev is development test set and test-std is standard test set for VQA 2.0.

| Model | Ens | VQA 2.0 val | | | | VQA 2.0 test-dev | | | | VQA 2.0 test-std | | | |
|---|---|---|---|---|---|---|---|---|---|---|---|---|---|
| | | Yes/No | Number | Other | All | Yes/No | Number | Other | All | Yes/No | Number | Other | All |
| Andreas et al. (2016) | - | 73.38 | 33.23 | 39.93 | 51.62 | - | - | - | - | - | - | - | - |
| Yang et al. (2016) | - | 68.89 | 34.55 | 43.80 | 52.20 | - | - | - | - | - | - | - | - |
| Teney et al. (2018) | - | 80.07 | 42.87 | 55.81 | 63.15 | 81.82 | 44.21 | 56.05 | 65.32 | 82.20 | 43.90 | 56.26 | 65.67 |
| Teney et al. (2018) | ✓ | - | - | - | - | 86.08 | 48.99 | 60.80 | 69.87 | 86.60 | 48.64 | 61.15 | 70.34 |
| Yu et al. (2018) | - | - | - | - | - | 84.27 | 49.56 | 59.89 | 68.76 | - | - | - | - |
| Yu et al. (2018) | ✓ | - | - | - | - | - | - | - | - | 86.65 | 51.13 | 61.75 | 70.92 |
| Zhang et al. (2018) | - | - | 49.36 | - | 65.42 | 83.14 | 51.62 | 58.97 | 68.09 | 83.56 | 51.39 | 59.11 | 68.41 |
| Kim et al. (2018)* | - | - | - | - | 66.04 | 85.43 | 54.04 | 60.52 | 70.04 | 85.82 | 53.71 | 60.69 | 70.35 |
| Kim et al. (2018)* | ✓ | - | - | - | - | 86.68 | 54.94 | 62.08 | 71.40 | 87.22 | 54.37 | 62.45 | 71.84 |
| Jiang et al. (2018)* | ✓ | - | - | - | - | 87.82 | 51.54 | 63.41 | 72.12 | 87.82 | 51.59 | 63.43 | 72.25 |
| baseline $M_{\text{vqa}_0}$ | - | 80.28 | 43.06 | 53.21 | 62.05 | - | - | - | - | - | - | - | - |
| PMN $M_{\text{vqa}_5}$ | - | 82.48 | 48.15 | 55.53 | 64.68 | 84.07 | 52.12 | 57.99 | 68.07 | - | - | - | - |
| PMN $M_{\text{vqa}_5}$ | ✓ | - | - | - | - | 85.74 | 54.39 | 60.60 | 70.25 | 86.34 | 54.26 | 60.80 | 70.68 |

questions. This synthetic data helps to increase the accuracy by ∼1% for all module variants. Since the number of questions that have relational reasoning and counting (*e.g.* how many people are sitting on the sofa? how many plates on table?) is limited, we also sample relational synthetic questions from VG. These questions are used only to improve the parameters of query transmitter $Q_{\text{cnt}\rightarrow\text{rel}}$ for the relationship module. Table 2 shows a large improvement (4.6%) of the compositional module over the non-modular baseline. When training for the next task (VQA), unlike other modules whose parameters are fixed, we *fine-tune* the counting module because counting module expects the same form of input - embedding of natural language question. The performance of counting module depends crucially on the quality of attention map over bounding boxes. By employing more questions from the whole VQA dataset, we obtain a better attention map, and the performance of counting module increases from 50.0% (*c.f.* Table 2) to 55.8% with finetuning (see Appx B.2 for more details).

**Visual Question Answering.** We present ablation studies on the val set of VQA 2.0 in Table 3. As seen, PMN strongly benefits from utilizing different modules achieving a performance improvement of 2.6% over the baseline. Note that all results here are without additional questions from the VG data. We also compare performance of PMN for the VQA task with state-of-the-art models in Table 4. Models are trained on the train split for results on VQA val, while for test-dev and test-std, models are trained on both the train and val splits. Although we start with a much lower baseline performance of 62.05% on the val set (vs. 65.42% (Zhang et al., 2018), 63.15% (Teney et al., 2018), 66.04% (Kim et al., 2018)), PMN's performance is on par with these models. Note that entries with * are parallel works to ours. Also, as Jiang et al. (2018) showed, the performance depends strongly on engineering choices such as learning rate scheduling and ensembling models with different architectures.

**Plug-and-play architecture.** The query-answer communication within PMN results in a plug-and-play architecture where modules can be replaced by their improved versions. Instead of using generated captions from $M_{\text{cap}}$, when we feed in the ground-truth captions to *trained* $M_{\text{vqa}_5}$ in Table 3 (with 64.68% acc.), it achieves 65.43%. We also tried training and validating $M_{\text{vqa}_5}$ with ground-truth captions, and this achieved 67.84%. These results shows how PMN can be continually improved.

**Three additional experiments on VQA.** (1) To verify that the gain is not from the increased model capacity, we trained a baseline with the number of parameters approximately matching that of the full PMN model. This baseline with more capacity also achieves 62.0%, thus confirming our claim. (2) We evaluated the impact of the additional data available. We convert the subj-obj-rel triplets used for the relationship detection task to additional QAs (e.g. Q: what is on top of the desk?, A: laptop) and train the $M_{\text{vqa}_1}$ model (Table 3). This results in an accuracy of 63.05%, not only lower than $M_{\text{vqa}_2}$ (63.64%) that uses the relationship module via PMN, but also lower than $M_{\text{vqa}_1}$ at 63.38%. This suggests that while additional data may change the question distribution and reduce performance, PMN is robust and benefits from a separate relationship module. (3) Lastly, we conducted another

experiment to show that PMN does make efficient use of the lower level modules. We give equal importance scores to all modules in $M_{\mathrm{vqa}_5}$ model (Table 3) (thus, fixed computation path), achieving 63.65% accuracy. While this is higher than the baseline at 62.05%, it is lower than $M_{\mathrm{vqa}_5}$ at 64.68% which softly chooses which sub-modules to use.

## 4.2 INTERPRETABILITY ANALYSIS

**Visualizing the model's reasoning process.** We present a qualitative analysis of the answering process. In Fig. 2, $M_{\mathrm{vqa}}$ makes query $q_{\mathrm{rel}} = [b_i, r]$ to $M_{\mathrm{rel}}$ where $b_i$ corresponds to the blue box 'bird' and $r$ corresponds to 'on top of' relationship. $M_{\mathrm{vqa}}$ correctly chooses (*i.e.* higher importance score) to use $M_{\mathrm{rel}}$ rather than its own output produced by $\Omega_{\mathrm{vqa}}$ since the question requires relational reasoning. With the attended green box obtained from $M_{\mathrm{rel}}$, $M_{\mathrm{vqa}}$ mostly uses the object and captioning modules to produce the final answer. More examples are presented in Appendix C.

**Judging Answering Quality.** The modular structure and gating mechanism of PMN makes it easy to interpret the reasoning behind the outputs. We conduct a human evaluation with Amazon Mechanical Turk on 1,600 randomly chosen questions. Each worker is asked to rate the explanation generated by the baseline model and the PMN like a teacher grades student exams in the scale of 0 (very bad), 1 (bad), 2 (satisfactory), 3 (good), 4 (very good). The baseline explanation is composed of the bounding box it attends to and the final answer. For PMN, we form a rule-based natural language explanation based on the prominent modules used. An example is shown in Fig. 3. Each question is assessed by three human workers.

Table 5: Average human judgments from 0 to 4. ✓ indicates that model got final answer right, and ✗ for wrong.

| Correct? | | #Q | Human Rating | |
|---|---|---|---|---|
| PMN | Baseline | | PMN | Baseline |
| ✓ | ✓ | 715 | 3.13 | 2.86 |
| ✓ | ✗ | 584 | 2.78 | 1.40 |
| ✗ | ✓ | 162 | 1.73 | 2.47 |
| ✗ | ✗ | 139 | 1.95 | 1.66 |
| All images | | 1600 | 2.54 | 2.24 |

Incorrect reasoning steps are penalized, so if PMN produces wrong reasoning steps, it could get a low score. On the other hand, the baseline model often scores well on simple questions that do not need complex reasoning (*e.g.* what color is the cat?).

We report results in Table 5, and show more examples in Appendix D. Human evaluators tend to give low scores to wrong answers and high scores to correct answers regardless of explanations, but PMN always scores higher if both PMN and baseline gets a question correct or wrong. Interestingly, a correct answer from PMN gets 1.38 points higher than wrong baseline, but a correct baseline scores only 0.74 higher than a wrong PMN answer. This shows that PMN gets partial marks even when it gets an answer wrong since the reasoning steps are partially correct.

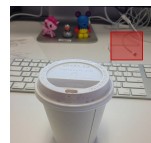

**Q: what is behind the men?**
• I first find the **BLUE** box, and then from that, I look at the **GREEN** box.
• The object **'tree'** would be useful in answering the question.
• In conclusion, I think the answer is **trees**.

**Q: what color is the curvy wire?**
• I look at the **RED** box.
• The object properties **white long electrical** would be useful in answering the question.
• In conclusion, I think the answer is **white**.

**Low Data Regime.** PMN benefits from re-using modules and only needs to learn the communication between them. This allows us to achieve good performance even when using a fraction of the training data. Table 6 presents the absolute gain in accuracy PMN achieves. For this experiment, we use $\mathcal{L}_{\mathrm{vqa}} =$

Figure 3: Example of PMN's reasoning processes. **Top**: it correctly first find a person and then uses relationship module to find the tree behind him. **Bottom**: it finds the wire and then use attribute module to correctly infer its attributes - white, long, electrical - and then outputs the correct answer.

$[\Omega_{vqa}, M_{\mathrm{rel}}, M_{\mathrm{obj}}, M_{\mathrm{att}}, \Delta_{vqa}, M_{\mathrm{cap}}]$ (because of overlapping questions from $M_{\mathrm{cnt}}$). When the amount of data is really small (1%), PMN does not help because there is not enough data to learn to communicate with lower modules. The maximum gain is obtained when using 10% of data. It shows that PMN can help in situations where there is not a huge amount of training data since it can exploit previously learned knowledge. The gain remains constant at about 2% from then on.

Table 6: Absolute gain in accuracy when using a fraction of the training data.

| Fraction of VQA training data (in %) | 1 | 5 | 10 | 25 | 50 | 100 |
|---|---|---|---|---|---|---|
| Absolute accuracy gain (in %) | -0.49 | 2.21 | 4.01 | 2.66 | 1.79 | 2.04 |

## 5 CONCLUSION AND DISCUSSION

In this work, we proposed Progressive Module Networks (PMN) that train task modules in a compositional manner, by exploiting previously learned lower-level task modules. PMN can produce queries to call other modules and make use of the returned information to solve the current task. Given experts in specific tasks, the parent module only needs to learn how to effectively communicate with them. It can also choose which lower level modules to use. Thus, PMN is data efficient and

provides a more interpretable reasoning processes. Also, since there is no need to know about the inner workings of the children modules, it opens up promising ways to train intelligent robots that can compartmentalize and perform multiple tasks as they progressively improve their abilities. Moreover, one task can benefit from unrelated tasks unlike conventional multi-task learning algorithms.

PMN as it stands has few limitations with respect to hand-designed structures and the need for additional supervision. Nevertheless, PMN is an important step towards more interpretable, compositional multi-task models. Some of the questions to be solved in the future include: 1) learning module lists automatically; 2) choosing few modules (hard attention) to reduce overhead; 3) more generic structure of module components across tasks; and 4) joint training of all modules.

**Acknowledgments.** Partially supported by the DARPA Explainable AI (XAI) program, Samsung and NSERC. We also thank NVIDIA for their donation of GPUs.

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

APPENDICES

# A    MODULE ARCHITECTURES

We discuss the detailed architecure of each module. We first describe the shared environment and soft attention mechanism architecture.

**Environment.**    The sensory input that form our environment $\mathcal{E}$ consists of: **(i)** *image regions*: $N$ image regions $X = [X_1, \ldots, X_N]$, each $X_i \in \mathbb{R}^d$ with corresponding bounding box coordinates $\mathbf{b} = [b_1, \ldots, b_N]$ extracted from Faster R-CNN (Ren et al., 2015); and **(ii)** *language*: vector representation of a sentence $S$ (in our example, a question). $S$ is computed through a one layer GRU by feeding in the embedding of each word $[w_1, \ldots, w_T]$ at each time step. For (i), we use a pretrained model from Anderson et al. (2018) to extract features and bounding boxes.

**Soft attention.**    For all parts that use soft-attention mechanism, an MLP is emloyed. Given some *key vector* $k$ and a set of data to be attended $\{d_1, \ldots, d_N\}$, we compute

$$\text{attention\_map} = (z(f(k) \cdot g(d_1)), \ldots, z(f(k) \cdot g(d_N))) \tag{1}$$

where $f$ and $g$ are a sequence of linear layer followed by ReLU activation function that project $k$ and $d_i$ into the same dimension, and $z$ is a linear layer that projects the joint representation into a single number. Note that we do not specify softmax function here because sigmoid is used for some cases.

## A.1    OBJECT AND ATTRIBUTE CLASSIFICATION (LEVEL 0)

The input to both modules $M_{\text{obj}}, M_{\text{att}}$ is a visual descriptor for a bounding box $b_i$ in the image, *i.e.*, $q_{\text{obj}} = X_i$. $M_{\text{obj}}$ and $M_{\text{att}}$ projects the visual feature $X_i$ to a 300-dimensional vector through a single layer neural network followed by $\tanh()$ non-linearity. We expect this vector to represent the name and attributes of the box $b_i$.

## A.2    IMAGE CAPTIONING (LEVEL 1)

$M_{\text{cap}}$ takes zero vector as the model input and produces natural language sentence as the output based on the environment $\mathcal{E}$ (detected image regions in an image). It has $\mathcal{L}_{\text{cap}} = [\Omega_{\text{cap}}, M_{\text{obj}}, M_{\text{att}}, \Delta_{\text{cap}}]$ and goes through maximum of $T_{\text{cap}} = 16$ time steps or until it reaches the end of sentence token. $M_{\text{cap}}$ is implemented similarly to the captioning model in Anderson et al. (2018). We employ two layered GRU (Cho et al., 2014) as the recurrent state update function $U_{\text{cap}}$ where $s^t = (h_1^t, h_2^t)$ with hidden states of the first and second layers of $U_{\text{cap}}$. Each layer has 1000-d hidden states.

The state initializer $I_{\text{cap}}$ sets the initial hidden state of $U_{\text{cap}}$, or the model state $s^t$, as a zero vector. For $t$ in $T_{\text{cap}} = 16$, $M_{\text{cap}}$ does the following four operations:

(1) The importance function $G_{\text{cap}}$ is executed. It is implemented as a linear layer $\mathbb{R}^{1000} \to \mathbb{R}^4$ (for the four modules in $\mathcal{L}_{\text{cap}}$) that takes $s^t$, specifically $h_1^t \in s^t$ as input.

(2) $Q_{\text{cap} \to \Omega}$ passes $h_1^t$ to the attention module $\Omega_{\text{cap}}$ which attends over the image regions $X$ with $h_1^t$ as the key vector. $\Omega_{\text{cap}}$ is implemented as a soft-attention mechanism so that it produces attention probabilities $p_i$ (via softmax) for each image feature $X_i \in \mathcal{E}$. The returned attention map $v_\Omega$ is added to the scratch pad $V$.

(3) $Q_{\text{cap} \to \text{obj}}$ and $Q_{\text{cap} \to \text{att}}$ pass the sum of visual features $X$ weighted by $v_\Omega \in V$ to the corresponding modules. $\Delta_{\text{cap}}$ is implemented as an MLP. The receivers project the outputs into 1000 dimensional vectors $v_{\text{obj}}$, $v_{\text{att}}$, and $v_\Delta$ through a sequence of linear layers, batch norm, and $\tanh()$ nonlinearities. They are added to $V$.

(4) As stated above, $U_{\text{cap}}$ is a two-layered GRU. At time $t$, the first layer takes input the average visual features from the environment $\mathcal{E}$, $\frac{1}{N} \sum_i X_i$, embedding vector of previous word $w_{t-1}$, and $h_2^t$. For time $t = 1$, *beginning-of-sentence* embedding and zero vector are inputs for $w_1$ and $h_1^1$, respectively. The second layer is fed $h_1^t$ as well as the information from other modules,

$$\rho = \sum (\text{softmax}(g_{\text{obj}}, g_{\text{att}}, g_\Delta) \cdot (v_{\text{obj}}, v_{\text{att}}, v_\Delta)) \tag{2}$$

which is a gated summation of outputs in $V$ with softmaxed importance scores. We now have a new state $s^{t+1} = (h_1^{t+1}, h_2^{t+1})$.

The output of $M_{\mathrm{cap}}$, $o_{\mathrm{cap}}$, is a sequence of words produced through $\Psi_{\mathrm{cap}}$ which is a linear layer projecting each $h_2^t$ in $s^t$ to the output word vocabulary.

## A.3 RELATIONSHIP DETECTION (LEVEL 1)

Relationship detection task requires one to produce triplets in the form of "subject - relationship - object" (Lu et al., 2016). We re-purpose this task as one that involves finding the relevant item (region) in an image that is related to a given input through a given relationship. The input to the module is $q_{\mathrm{rel}} = [b_i, r]$ where $b_i$ is a one-hot encoded input bounding box (whose $i$-th entry is 1 and others 0) and $r$ is a one-hot encoded relationship category (*e.g.* above, behind). $M_{\mathrm{rel}}$ has $\mathcal{L}_{\mathrm{rel}} = [M_{\mathrm{obj}}, M_{\mathrm{att}}, \Delta_{\mathrm{rel}}]$ and goes through $T_{rel} = N$ steps where $N$ is the number of bounding boxes (image regions in the environment). So at time step $t$, the module looks at the $t$-th box. $M_{\mathrm{rel}}$ uses $M_{\mathrm{obj}}$ and $M_{\mathrm{att}}$ just as feature extractors for each bounding box. Therefore, it does not have a complex structure.

The state initializer $I_{\mathrm{rel}}$ projects $r$ to a 512 dimensional vector with an embedding layer, and the resulting vector is set as the first state $s^1$.

For $t$ in $T_{\mathrm{rel}} = N$, $M_{\mathrm{rel}}$ does the following three operations:

(1) $Q_{\mathrm{rel}\to\mathrm{obj}}$ and $Q_{\mathrm{rel}\to\mathrm{att}}$ pass the image vector corresponding to the bounding box $b_t$ to $M_{\mathrm{obj}}$ and $M_{\mathrm{att}}$. $R_{\mathrm{obj}\to\mathrm{rel}}$ and $R_{\mathrm{att}\to\mathrm{rel}}$ are identity functions, *i.e.*, we do not modify the object and attribute vectors. The outputs $v_{\mathrm{obj}}$ and $v_{\mathrm{att}}$ are added to $V$.

(2) $\Delta_{\mathrm{rel}}$ projects the coordinate of the current box $b_t$ to a 512 dimensional vector. This resulting $v_{\Delta}$ is added to $V$.

(3) $U_{\mathrm{rel}}$ concatenates the visual feature $X_t$ with $v_{\mathrm{obj}}, v_{\mathrm{att}}, v_{\Delta}$ from $V$. The concatenated vector is fed through a MLP resulting in 512 dimensional vector. This corresponds to the new state $s^{t+1}$.

After $N$ steps, the prediction function $\Psi_{\mathrm{rel}}$ does the following operations:
The first state $s^1$ which contains relationship information is multiplied element-wise with $s^{i+1}$ (Note: $s^{i+1}$ corresponds to the input box $b_i$). Let such a vector be $l$. It produces an attention map $b_{\mathrm{out}}$ over all bounding boxes in $b$. The inputs to the attention function are $s^2, \ldots, s^{T_{\mathrm{rel}}}$ (i.e. all image regions) and the key vector $l$. $o_{\mathrm{rel}} = b_{\mathrm{out}}$ is the output of $M_{\mathrm{rel}}$ which represents an attention map indicating the bounding box that contains the related object.

## A.4 COUNTING (LEVEL 2)

Given a vector representation of a natural language question (*e.g.* how many cats are in this image?), the goal of this module is to produce a count. The input $q_{\mathrm{cnt}} = S \in \mathcal{E}$ is a vector representing a natural language question. When training $M_{\mathrm{cnt}}$, $q_{\mathrm{cnt}}$ is computed through a one layer GRU with hidden size of 512 dimensions. The input to the GRU at each time step is the embedding of each word from the question. Word embeddings are initialized with 300 dimensional GloVe word vectors (Pennington et al., 2014) and fine-tuned thereafter. Similar to visual features obtained through CNN, the question vector is treated as an environment variable. $M_{\mathrm{cnt}}$ has $\mathcal{L}_{\mathrm{cnt}} = [\Omega_{\mathrm{cnt}}, M_{\mathrm{rel}}]$ and goes through only one time step.

The state initializer $I_{\mathrm{cnt}}$ is a simple function that just sets $s^1 = q_{\mathrm{cnt}}$.

For $t$ in $T_{\mathrm{cnt}} = 1$, $M_{\mathrm{cnt}}$ does the following four operations:

(1) The importance function $G_{\mathrm{cnt}}$ is executed. It is implemented as a linear layer $\mathbb{R}^{512} \to \mathbb{R}^2$ (for the two modules in $\mathcal{L}_{\mathrm{cnt}}$) that takes $s^t$ as input.

(2) $Q_{\mathrm{cnt}\to\Omega}$ passes $s^t$ to the attention module $\Omega_{\mathrm{cnt}}$ which attends over the image regions $X$ with $s^t$ as the key vector. $\Omega_{\mathrm{cnt}}$ is implemented as an MLP that computes a dot-product soft-attention similar to Yang et al. (2016). The returned attention map $v_{\Omega}$ is added to the scratch pad $V$.

(3) $Q_{\mathrm{cnt}\to\mathrm{rel}}$ produces an input tuple $[b, r]$ for $M_{\mathrm{rel}}$. The input object box $b$ is produced by a MLP that does soft attention on image boxes, and the relationship category $r$ is produced through a MLP with $s^t$ as input. $M_{\mathrm{rel}}$ is called with $[b, r]$ and the returned map $v_{\mathrm{rel}}$ is added to $V$.

(4) $U_{\text{cnt}}$ first computes probabilities of using $v_\Omega$ or $v_{\text{rel}}$ by doing a softmax over the importance scores. $v_\Omega$ and $v_{\text{rel}}$ are weighted and summed with the softmax probabilities resulting in the new state $s^2$ containing the attention map. Thus, the state update function chooses the map from $M_{\text{rel}}$ if the given question involves in relational reasoning.

The prediction function $\Psi_{\text{cnt}}$ returns a count vector. The count vector is computed through the counting algorithm by Zhang et al. (2018), which builds a graph representation from attention maps to count objects. The method uses $s^2$ through a sigmoid and bounding box coordinates $b$ as inputs. Zhang et al. (2018) is a fully differentiable algorithm and the resulting count vector corresponds to one-hot encoding of a number. We let the range of count be 0 to $12 \in \mathbb{Z}$. Please refer to Zhang et al. (2018) for details of the counting algorithm.

## A.5 Visual Question Answering (Level 3)

The description for the VQA task (Sec. 3.4) is included here again for completeness. The input $q_{\text{vqa}}$ is a vector representing a natural language question (*i.e.* the sentence vector $S \in \mathcal{E}$). The state variable $s^t$ is represented by a tuple $(q_{\text{vqa}}^t, k^{t-1})$ where $q_{\text{vqa}}^t$ represents query to ask at time $t$ and $k^{t-1}$ represents knowledge gathered at time $t-1$. The state initializer $I_{\text{vqa}}$ is composed of a GRU with hidden state dimension 512. The first input to GRU is $q_{\text{vqa}}$, and $I_{\text{vqa}}$ sets $s^1 = (q_{\text{vqa}}^1, \mathbf{0})$ where $q_{\text{vqa}}^1$ is the first hidden state of the GRU and $\mathbf{0}$ is a zero vector (no knowledge at first).

For $t$ in $T_{\text{vqa}} = 2$, $M_{\text{vqa}}$ does the following seven operations:

(1) The importance function $G_{\text{vqa}}$ is executed. It is implemented as a linear layer $\mathbb{R}^{512} \to \mathbb{R}^7$ (for the seven modules in $\mathcal{L}_{\text{vqa}}$) that takes $s^t$, specifically $q_{\text{vqa}}^t \in s^t$ as input.

(2) $Q_{\text{vqa} \to \Omega}$ passes $q_{\text{vqa}}^t$ to the attention module $\Omega_{\text{vqa}}$ which attends over the image regions $X$ with $q_{\text{vqa}}^t$ as the key vector. $\Omega_{\text{vqa}}$ is implemented as an MLP that computes a dot-product soft-attention similar to Yang et al. (2016). The returned attention map $v_\Omega$ is added to the scratch pad $V$.

(3) $Q_{\text{vqa} \to \text{rel}}$ produces an input tuple $[b, r]$ for $M_{\text{rel}}$. The input object box $b$ is produced by a MLP that does soft attention on image boxes, and the relationship category $r$ is produced through a MLP with $q_{\text{vqa}}^t$ as input. $M_{\text{rel}}$ is called with $[b, r]$ and the returned map $v_{\text{rel}}$ is added to $V$.

(4) $Q_{\text{vqa} \to \text{obj}}$, $Q_{\text{vqa} \to \text{att}}$, and $Q_{\text{vqa} \to \Delta}$ first compute a joint attention map $m$ as summation of $(v_\Omega, v_{\text{rel}})$ weighted by the softmaxed importance scores of $(\Omega_{\text{vqa}}, M_{\text{rel}})$, and they pass the sum of visual features $X$ weighted by $m$ to the corresponding modules. $\Delta_{\text{vqa}}$ is implemented as an MLP. The receivers project the outputs into 512 dimensional vectors $v_{\text{obj}}$, $v_{\text{att}}$, and $v_\Delta$ through a sequence of linear layers, batch norm, and $\tanh()$ nonlinearities. They are added to $V$.

(5) $Q_{\text{vqa} \to \text{cnt}}$ passes $q_{\text{vqa}}^t$ to $M_{\text{cnt}}$ which returns $o_{\text{cnt}}$. $R_{\text{cnt} \to \text{vqa}}$ projects the count vector $o_{\text{cnt}}$ into a 512 dimensional vector $v_{cnt}$ through the same sequence of layers as above. $v_{\text{cnt}}$ is added to $V$.

(6) $M_{\text{vqa}}$ calls $M_{\text{cap}}$ and $R_{\text{cap} \to \text{vqa}}$ receives natural language caption of the image. It converts words in the caption into vectors $[w_1, \ldots, w_T]$ through an embedding layer. The embedding layer is initialized with 300 dimensional GloVe vectors (Pennington et al., 2014) and fine-tuned. It does softmax attention operation over $[w_1, \ldots, w_T]$ through a MLP with $q_{\text{vqa}}^t \in s_t$ as the key vector, resulting in word probabilities $p_1, \ldots, p_T$. The sentence representation $\sum_i^T p_i \cdot w_i$ is projected into a 512 dimensional vector $v_{\text{cap}}$ using the same sequence as $v_{\text{cnt}}$. $v_{\text{cap}}$ is added to $V$.

(7) The state update function $U_{\text{vqa}}$ first does softmax operation over the importance scores of $(M_{\text{obj}}, M_{\text{att}}, \Delta_{\text{vqa}}, M_{\text{cnt}}, M_{\text{cap}})$. We define an intermediate knowledge vector $k^t$ as the summation of $(v_{\text{obj}}, v_{\text{att}}, \delta_{\text{vqa}}, v_{\text{cnt}}, v_{\text{cap}})$ weighted by the softmaxed importance scores. $U_{\text{vqa}}$ passes $k^t$ as input to the GRU initialized by $I_{\text{vqa}}$, and we get $q_{\text{vqa}}^{t+1}$ the new hidden state of the GRU. The new state $s^{t+1}$ is set to $(q_{\text{vqa}}^{t+1}, k^t)$. This process allows the GRU to compute new question and state vectors based on what has been *asked* and *seen*.

After $T_{\text{vqa}}$ steps, the prediction function $\Psi_{\text{vqa}}$ computes the final output based on the initial question vector $q_{\text{vqa}}$ and all knowledge vectors $k^t \in s^t$. Here, $q_{\text{vqa}}$ and $k^t$ are fused with gated-tanh layers and fed through a final classification layer similar to Anderson et al. (2018), and the logits for all time steps are added. The resulting logit is the final output $o_{\text{vqa}}$ that corresponds to an answer in the vocabulary of the VQA task.

## B    ADDITIONAL EXPERIMENTAL DETAILS

In this section, we provide more details about datasets and module training.

### B.1    DATASETS

We extract bounding boxes and their visual representations using a pretrained model from Anderson et al. (2018)which is a Faster-RCNN (Ren et al., 2015) based on ResNet-101 (He et al., 2016). It produces 10 to 100 boxes with 2048-d feature vectors for each region. To accelerate training, we remove overlapping bounding boxes that are most likely duplicates (area overlap IoU > 0.7) and keep only the 36 most confident boxes (when available).

**MS-COCO** contains ∼100K images with annotated bounding boxes and captions. It is a widely used dataset used for benchmarking several vision tasks such as captioning and object detection.

**Visual Genome** is collected to relate image concepts to image regions. It has over 108K images with annotated bounding boxes containing 1.7M visual question answering pairs, 3.8M object instances, 2.8M attributes and 1.5M relationships between two boxes. Since the dataset contains MS-COCO images, we ensure that we do not train on any MS-COCO validation or test images.

**VQA 2.0** is the most popular visual question-answering dataset, with 1M questions on 200K natural images. Questions in this dataset require reasoning about objects, actions, attributes, spatial relations, counting, and other inferred properties; making it an ideal dataset for our visual-reasoning PMN.

### B.2    TRAINING

Here, we give training details of each module. We train our modules sequentially, from low level to high level tasks, one at a time. When training a higher level module, internal weights of the lower level modules are not updated, thus preserving their performance on the original task. We do train the weights of the residual module $\Delta$ and the attention module $\Omega$. We train $I$, $G$, $Q$, $R$, $U$, and $\Psi$, by allowing gradients to pass through the lower level modules. Thus, while the existing lower modules are held fixed, the new module learns to communicate with them via the query transmitter $Q$ and receiver $R$.

**Object and attribute classification.** $M_{\mathrm{obj}}$ is trained to minimize the cross-entropy loss for predicting object class by including an additional linear layer on top of the module output. $M_{\mathrm{att}}$ also include an additional linear layer on top of the module output, and is trained to minimize the binary cross-entropy loss for predicting attribute classes since one detected image region can contain zero or more attribute classes. We make use of 780K/195K train/val object instances paired with attributes from the Visual Genome dataset. They are trained with the Adam optimizer at learning rate of 0.0005 with batch size 32 for 20 epochs.

**Image captioning.** $M_{\mathrm{cap}}$ is trained using cross-entropy loss at each time step (maximum likelihood). Parameters are updated using the Adam optimizer at learning rate of 0.0005 with batch size 64 for 20 epochs. We use the standard split of MS-COCO captioning dataset.

**Relationship detection.** $M_{\mathrm{rel}}$ is trained using cross-entropy loss on "subject - relationship - object" pairs with Adam optimizer with learning rate of 0.0005 with batch size 128 for 20 epochs. The pairs are extracted from the Visual Genome dataset that have both subject and object boxes overlapping with the ground truth boxes (IoU > 0.7), resulting in 200K/38K train/val tuples.

**Counting.** $M_{\mathrm{cnt}}$ is trained using cross-entropy loss on questions starting with 'how many' from the VQA 2.0 dataset. We use Adam optimizer with learning rate of 0.0001 with batch size 128 for 20 epochs. As stated in the experiments section, we additionally create ∼89K synthetic questions to increase our training set by counting the object boxes and forming 'how many' questions from the VG dataset (*e.g.* (Q: how many dogs are in this picture?, A:3) from an image containing three bounding boxes of dog). We also sample relational synthetic questions from each training image from VG that are used to train only the module communication parameters when the relationship module is included. We use the same 200K/38K split from the relationship detection task by concatenating 'how many'+subject+relationship' or 'how many'+relationship+object (*e.g.* how many plates on table?, how many behind door?). The module communication parameters for $M_{\mathrm{rel}}$ in this case are $Q_{\mathrm{cnt}\rightarrow\mathrm{rel}}$

which compute a relationship category and the input image region to be passed to $M_{\rm rel}$. To be clear, we supervise $q_{\rm rel} = [b_i, r]$ to be sent to $M_{\rm rel}$ by reducing cross entropy loss on $b_i$ and $r$.

**Visual Question Answering.** $M_{\rm vqa}$ is trained using binary cross-entropy loss on $o_{\rm vqa}$ with Adam optimizer with learning rate of 0.0005 with batch size 128 for 7 epochs. We empirically found binary cross-entropy loss to work better than cross-entropy which was also reported by Anderson et al. (2018). Unlike other modules whose parameters are fixed, we *fine-tune* only the counting module because counting module expects the same form of input - embedding of natural language question. The performance of counting module depends crucially on the quality of attention map over bounding boxes. By employing more questions from the whole VQA dataset, we obtain a better attention map, and the performance of counting module increases from 50.0% to 55.8% with finetuning. Since $M_{\rm vqa}$ and $M_{\rm cnt}$ expect the same form of input, the weights of attention modules $\Omega_{\{\rm vqa,cnt\}}$ and query transmitters for the relationship module $Q_{\{\rm vqa,cnt\}\rightarrow\rm rel}$ are shared.

## C  PMN EXECUTION ILLUSTRATED

We provide more examples of the execution traces of PMN on the visual question answering task in Figure 4. Each row in the figure corresponds to different examples. For each row in the figure, the left column shows the environment $\mathcal{E}$, the middle column shows the final answer & visualizes step 3 in the execution process, and the right column shows computed importance scores along with populated scratch pad.

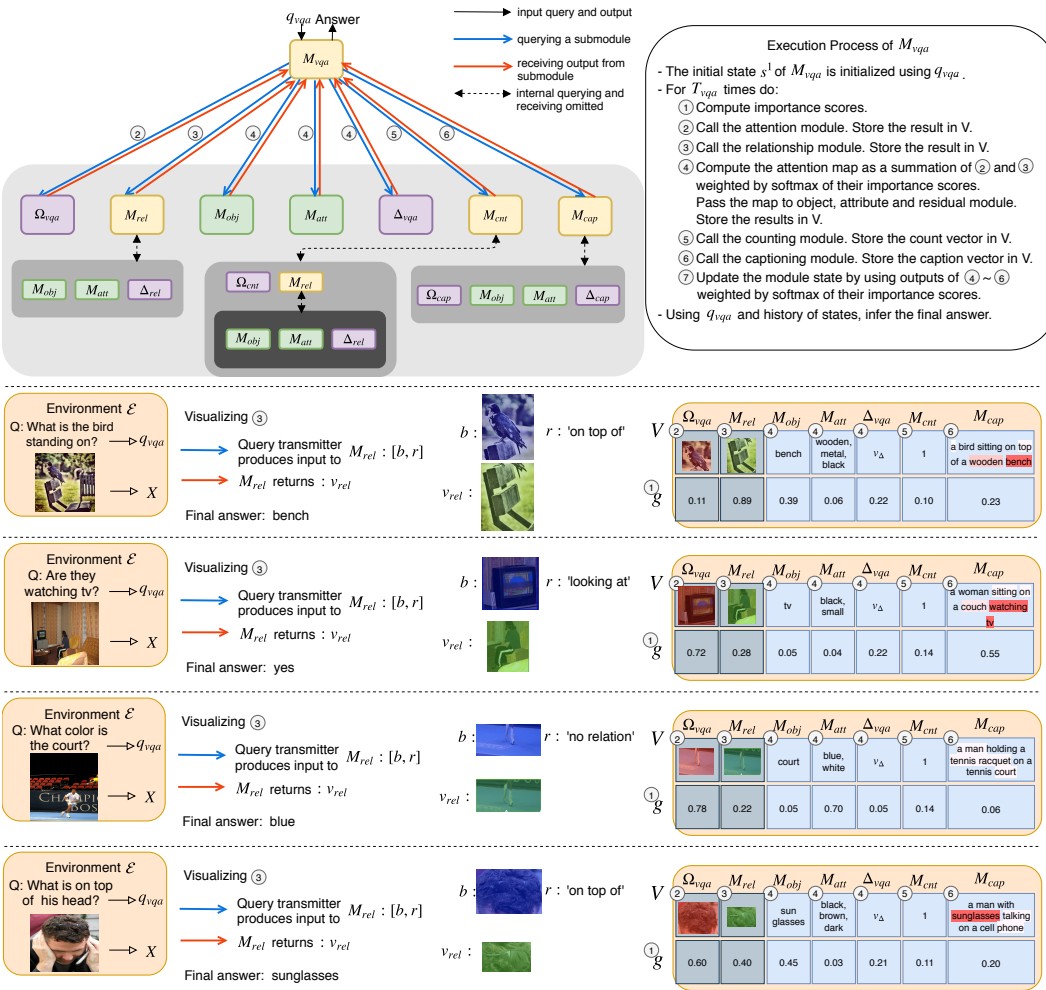

Figure 4: Example of PMN's module execution trace on the VQA task. Numbers in circles indicate the order of execution. Intensity of gray blocks represents depth of module calls. All variables including queries and outputs stored in $V$ are vectorized to allow gradients to flow (*e.g.*, caption is composed of a sequence of softmaxed $W$ dimensional vectors for vocabulary size $W$). For $M_{\text{cap}}$, words with higher intensity in red are deemed more relevant by $R^{\text{cap}}_{\text{vqa}}$.

# D    EXAMPLES OF PMN'S REASONING

We provide more examples of the human evaluation experiment on interpretability of PMN compared with the baseline model in Figure 5.

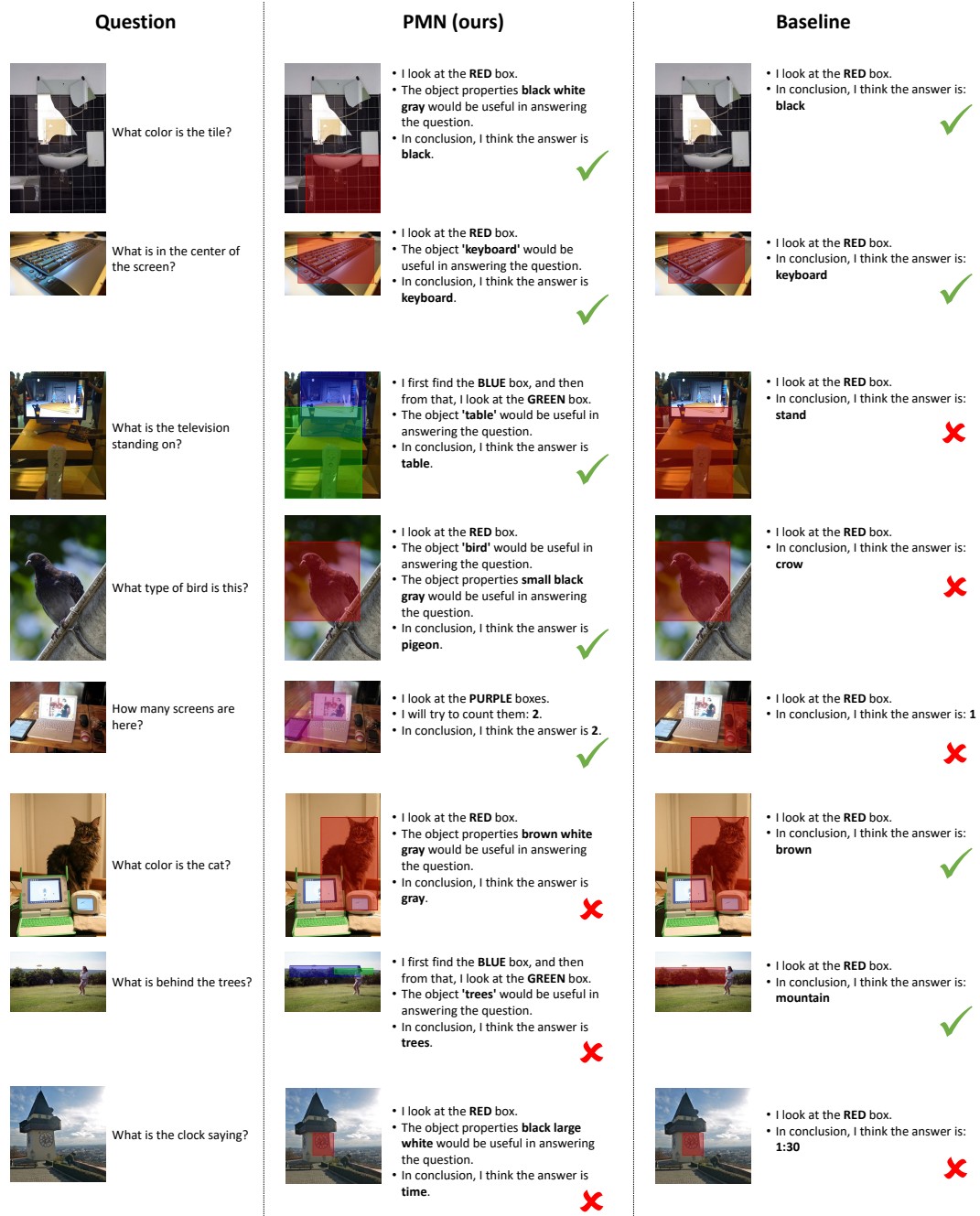

Figure 5: Example of PMN's reasoning processes compared with the baseline given the question on the left. ✓ and ✗ denote correct and wrong answers, respectively.

