# OpenReview forum: "Visual Reasoning by Progressive Module Networks"
_ICLR.cc/2019/Conference_

### Official Review · AnonReviewer1 · 2018-11-01

**Rating:** 6
**Confidence:** 4

**Review:**

Summary:
The authors propose a network for VQA incorporating hand-crafted modules and their hierarchy, each of which is a network for a high-level vision task. Some modules may share the same sub-modules at a different level in the module hierarchy. Each module is individually (not end-to-end) trained with a dataset containing a dedicated annotation for their high-level tasks. The proposed model shows comparable scores to the existing models.

Presentation and clarity:
The paper is well written and easy to follow and contains reasonable experiments for understanding the proposed method.

Originality and significance:
I mainly do not agree that this work generalizes NMN. Instead, I believe that this work is a special case of NMN where the modules and their hierarchy are manually defined based on the authors' intuition. Meanwhile, the proposed network architecture is static, and thus the main idea of having multiple modules in a network is not novel as other approaches using static network architectures such as [A] also facilitate multiple modules for different sub-procedures (e.g., RNN for questions and CNN for image) and sometimes share modules in multiple stages too. The main difference between this and previous works is that the modules in this work deal with high-level tasks chosen by the authors. I am not convinced that designing the modules with high-level tasks is a better choice over designing modules that are less task-specific. Rather, I see more drawbacks as the proposed method requires multiple datasets with diverse task-specific annotation. Also, the modules and their connectivity are less scalable and extendable as they are not learned.

Considering all the model and dataset complexities, the improvements over black-box models are mostly marginal. The main benefits we get from all these complexities are the interpretability. However, for many modules, the interpretability comes from indirect signals that are often not clear how to interpret for the question answering. On the other hand, the manually designed sub-tasks may cause error propagation in the network as these modules are not directly optimized for the final objective.

Some questions and comments:
I do not understand why it is necessary to have the image captioning module as it does not directly relate to the question answering. Moreover, the caption itself is generated without conditioning on the question.

[A] Yang, Zichao, et al. "Stacked attention networks for image question answering." CVPR 2016.


== After discussion phase
Based on the rebuttal and additional experiments that clarified and resolved my questions, I change my initial rating.

---

> ### Author Response · Authors · 2018-11-11
> **Response to Reviewer 1**
>
> Thanks for the feedback. We hope to convince you that PMN is a framework to learn continuously from previous knowledge, and not just a solution to VQA.
>
>
> 1. PMN vs. NMN
> - We agree that PMN is not a generalization of NMN. However, we argue that it is not a special case of NMN either. We highlight two significant differences:
>
> a) Progression: PMN is a framework that learns to do (visual) reasoning by starting with simpler tasks (object labels) and building up to more complex tasks (VQA). This is an important step towards building intelligent agents that continuously learn new tasks by using the tasks they are already good at. There is no sense of progression in NMN and everything is learned from scratch. From the experiments related to the low data regime (Table 6), we see that PMN can make efficient use of available data to learn to communicate with experts and solve the task.
>
> b) Task Modules: Communication in PMN is at the *query-answer level*. Since module outputs are answers to other (human-designed) tasks, the process is easier to interpret (more human-readable). On the other hand, NMN’s modules, as showcased in their paper, contained one or two conv. or linear layers and solve sub-functions such as attention or classification.
>
> We edited the paper to remove misunderstanding our wording may have caused.
>
>
> 2. PMN vs. Static models
> - In addition to the above differences to NMN, PMN has three more significant differences to static models ([A]).
>
> a) Dynamic choice of modules: PMN’s state and importance function choose which modules to consider. This can go even further, and using a threshold, we may not execute some modules at all (during inference). Static models always go through the same steps.
>
> b) Information propagates in a tree-like fashion: A high-level module asks for some information from a lower module, that further produces queries for its own lower modules (see Fig. 1). For example, VQA calls counting which calls relationship detection.
>
> c) Direct querying of lower modules: PMN produces explicit queries for lower tasks using the query transmitter Q (see Fig. 2, step 3). Based on the current state, it can choose to ask information about a specific query that may be helpful to answer the question.
>
> The inter-module communication and the computation graph are all learned.
>
>
> 3. Modules with high-level tasks, Multiple datasets
> - Task-specific models are the default practice in machine learning. However, to have an intelligent agent that can learn a host of tasks over time it is beneficial to have the tasks build on top of each other (See 1. (a)). This is similar to the human learning process where kids first learn object names and attributes, followed by increasingly harder tasks such as counting. Datasets in the community are typically focused on one specific task, and thus we are forced to use multiple datasets and annotations to progressively learn visual reasoning abilities.
>
>
> 4. Minor improvements over black-box models, Interpretability
> -  We encourage the reviewer to look at our paper in a more holistic view. Our main aim is to mimic challenging real scenarios in which we want to train agents to learn many tasks, increasing in their complexity, rather than squeezing numbers for one particular dataset. The VQA dataset has a strong bias that is exploited by black-box models [B]. This is one of the reasons why PMN performs much better than the other models in the low data regime (Table 6) - the gap gets smaller with more data as black-box models learn to exploit dataset bias. The paper showcases how to learn tasks by progression and modularity. We hope this is interesting to readers beyond just the numbers. The fact that the performance also improves is a nice bonus.
>
> With respect to interpretability, the query-answer communication within PMN is more human-readable than other models (See 1. (b)). For example, as shown in Fig. 2, it produces queries for the relationship module (bird, ‘on top of’) and the relationship module returns the box corresponding to 'bench'. Other examples such as Fig. 3, App. C&D, and the human evaluation concretely support the fact that generated outputs are much more interpretable than standard attention maps.
>
>
> 5. Error propagation
> - PMN combines information from the lower modules through importance scores. For a given set of questions, if a module produces erroneous outputs, PMN can learn to ignore such outputs and rely on other modules or it’s own residual.
>
>
> 6. Captioning for VQA
> - In captioning, one describes the most salient aspects of the picture. For example, a caption “a married couple walking on the beach”, provides answers to several questions ('are they married?', 'where are they?', etc). If the actual question relates to these, then the VQA module can simply leverage the information. In response to R2, we evaluated how well VQA can leverage ground-truth captions and see a large 2.0% improvement.
>
>
> [B] Agrawal, et al. Overcoming Priors for VQA. 2018

---

> > ### Comment · AnonReviewer1 · 2018-11-25
> > **Thank you for your response**
> >
> > I appreciate the response from the reviewer.
> >
> > I accept the authors' arguments that PMNs are different from NMNs and static models, although I believe NMNs are proposed more generally and can still be designed with progression property. And, the explicit architecture design with the progression and the experiments on it should still be counted as the authors' contribution. Also, I also agree that the interpretability of the model is improved compared to previous methods.
> >
> > However, there are several things I still do not agree. To argue that it is beneficial to build a module for a task on top of other modules for lower-level tasks, the higher-level modules should show significantly improved performance compared to the other approaches. Otherwise, it can be thought that having an end-to-end black box model should be enough without the progression even though the concept of progression seems to be similar to the human learning process. To confirm this, I believe the improvements for intermediate tasks should be also measured since the learning process is "progressive".
> >
> > I agree that it is possible that the model has the capability to learn to avoid utilizing erroneous intermediate outputs. However, I am not sure if the model can correctly identify the erroneous outputs through unsupervised attention model. It is especially doubtful as the erroneous outputs are usually produced from hard examples. So the argument of model's capability for not utilizing erroneous outputs by the attention process should be experimentally verified.
> >
> > Overall, I agree that I underestimated some of the paper's contributions and thus want to raise my score. But, at the same time, I still see some weak points in the arguments that may be resolved by more experiments.

---

> > > ### Author Response · Authors · 2018-11-25
> > > **Response to the concerns**
> > >
> > > Thank you for the response. We address your concerns below.
> > >
> > >
> > > 1. Performance
> > >
> > > We believe the experiments section (Section 4.1), in particular Table 1 (relationship detection), 2 (counting) and 3 (visual question answering), shows how tasks benefit from utilizing submodules and learning progressively. Also, PMN for the VQA task achieves 64.68% validation accuracy which is an increase of more than 2.5% over the baseline (Table 3). This is without exploiting the additional questions from Visual Genome (which most other state-of-the-art models use, e.g. see Teney et al. 2018) and we do not employ additional data augmentation (Jiang et al. 2018).  Since PMN is a general framework, we also do not use advanced VQA-specific techniques such as bilinear attention (Kim et al. 2018) or ensembling different architectures (Jiang et al. 2018).
> > >
> > >
> > >
> > >
> > > 2. Erroneous outputs
> > >
> > > We agree that PMN might not be perfect in avoiding erroneous outputs, just like how children make mistakes in reasoning leading to incorrect conclusions. The human evaluation (Section 4.2) measures the quality of intermediate reasoning process as PMN with incorrect explanations are penalized by human workers. As shown in Table 5, PMN gets more part marks than the baseline even when its final output is incorrect. This shows intermediate outputs are good.  Moreover, as stated in the previous response, PMN can be naturally used as a plug-and-play model (see the "plug-and-play architecture" added in the experiments (Section 4.1)). Therefore, PMN has a very promising way of improving itself by utilizing better and less erroneous submodules, unlike other models. The query-and-answer level communication within PMN also makes it possible to have human feedback.
> > >
> > >
> > >
> > > - Teney et al. Tips and Tricks for Visual Question Answering: Learnings from the 2017 Challenge,  2018
> > > - Kim et al. Bilinear attention networks, 2018
> > > - Jiang et al. Pythia v0. 1: the winning entry to the VQA challenge 2018, 2018

---

> > > > ### Comment · AnonReviewer1 · 2018-11-25
> > > > **Thanks for the response**
> > > >
> > > > 1. I missed those results at the time of my response and now agree that the progression property of PMNs actually improves the performances of higher-level tasks. In addition to that, I also accept that the plug-and-play architecture is beneficial for further improvement in the future.
> > > >
> > > > 2. As the authors argue, the human evaluation involves the quality of the intermediate outputs. But this measure involves many other factors and is hard to understand the effects of the erroneous outputs. I believe including this kind of experiments would be still beneficial.

---

> > > > > ### Author Response · Authors · 2018-11-26
> > > > > **Additional experiment**
> > > > >
> > > > > We agree that inspecting module outputs is beneficial. It is hard to directly identify intermediate erroneous outputs as we do not have ground truth labels, so we conducted a new experiment that indirectly measures how erroneous outputs affect the model performance.
> > > > >
> > > > > Before that, we would like to clarify PMN chooses which modules to use, not which outputs to use (see line 5 in Algorithm 1). In the previous response, we stated “PMN combines information from the lower modules through importance scores. For a given set of questions, if a module produces erroneous outputs, PMN can learn to ignore such outputs and rely on other modules or it’s own residual.”. This might have caused the confusion, and we apologize. We meant to say - if PMN learns that certain modules’ outputs are not useful for solving a particular type of questions, it can learn to ignore those modules (with low importance scores) for that type of questions.
> > > > >
> > > > > As there are more than 200K questions in the val set, we focus on questions starting with ‘what’ and divide questions into 10 types. This amounts to 74K questions. For this experiment, we choose one submodule Mobj (the object classifier) of Mvqa and analyze 1) the type of questions when this module is used/not used, and 2) how Mobj’s output affects the final performance.
> > > > >
> > > > > Note: we say Mobj is ‘used’ if its importance score is higher than any other module.
> > > > >
> > > > > 1) We show the number of questions Mobj is used/not used depending on the question types:
> > > > >
> > > > >              Q types      ||
> > > > > (eg. what+‘time’) || #Qs Mobj used    | #Qs Mobj not used
> > > > > ---------------------------------------------------------------------------------------
> > > > > color                       ||            102              |            23628
> > > > > kind                        ||          8264              |             1760
> > > > > time                       ||             15               |             1731
> > > > > sport                      ||          1435             |               5
> > > > > animal                   ||          1093             |              46
> > > > > is the number      ||            326             | .           1727
> > > > > is on, in the          ||           3903            |              14
> > > > > brand                    ||             896            |              43
> > > > > is, are                    ||          24025           |             931
> > > > > room                     ||             931            |               0
> > > > >
> > > > >
> > > > > This shows PMN correctly learns to use/not use Mobj for certain types of questions (e.g. kind, sport, animal uses Mobj while color, time, number does not).
> > > > >
> > > > > 2) To see the effect of erroneous outputs, we select 23K questions out of the 74K questions whose ground-truth answer is in the vocabulary of Mobj and Mobj is used. That is, it is more likely that Mobj’s output is useful to infer the final answer.
> > > > > Let gt_ans be the ground-truth answer and obj_ans be the output label of Mobj.
> > > > >
> > > > >                                            ||  final answer correct  |  final answer incorrect |  Total
> > > > > ------------------------------------------------------------------------------------------------------------------------
> > > > > #Qs Obj_ans == gt_ans  ||   A.    12315  (82%)       |  B.   2690  (18%)             |  15005 (100%)
> > > > > #Qs Obj_ans != gt_ans   ||   C.    3233    (42%)       |  D.   4531  (58%)             |  7764 (100%)
> > > > >
> > > > > The numbers in the above table correspond to the number of questions. For example, A is the number of questions where Mobj’s output is equal to the ground-truth answer, leading to the correct final answer.
> > > > >
> > > > > For a large number of questions (A. 54%), PMN behaves as expected with correct Mobj’s outputs leading to correct Mvqa’s outputs. As A > B and C < D, we can see that if Mobj produces the correct answer, it is more likely that the final answer of Mvqa is correct, and if Mobj produces an erroneous output, it is more likely that the final answer is incorrect. We would like to stress that this is a weakness of not only our model but any other deep learning model. If some part of a model’s computation path is erroneous, it is more likely that the model performance suffers.
> > > > >
> > > > >
> > > > > B and C show cases where even though Mobj’s output was correct/incorrect, the final answer is incorrect/correct. One reason could be other contributing modules. Since PMN’s selection process is ‘soft’ using softmax of importance scores, other modules could confuse Mvqa. This could be moderated in future works by employing a hard selection process with gumbel softmax or reinforcement learning. It is also interesting to see that A (82%) is much greater than B (18%) and the difference between C (42%) and D (58%) is not as large. It suggests that other contributing modules might be helpful in situations where Mobj’s output is incorrect.
> > > > >
> > > > > It is not easy to directly quantify the effect of erroneous outputs without ground-truth labels. We hope you find these experiments helpful, and if you have another experiment in mind to show this more clearly, we would be happy to do it.

---

> > > > > > ### Comment · AnonReviewer1 · 2018-11-27
> > > > > > **Thank you for all your efforts**
> > > > > >
> > > > > > I deeply appreciate all your efforts for the extra experiments. Those tables also show meaningful statistics. However, I think you can simply measure what portion of questions are answered using M_obj (probably with soft weight thresholded) when M_obj outputs correct/incorrect answers. I think this can simply show if the model actually learns to ignore modules when they produce incorrect outputs.
> > > > > >
> > > > > > Again, I thank the authors for these additional experiments and I also want to point out that the authors resolved my questions and concerns.

---

> > > > > > > ### Author Response · Authors · 2018-11-27
> > > > > > > **Thank you for your suggestions.**
> > > > > > >
> > > > > > > What you suggested above is also a good way of looking at it.
> > > > > > > We tested it out:
> > > > > > >
> > > > > > >                              ||   Mobj used            |        Mobj not used . |            Total
> > > > > > > ----------------------------------------------------------------------------------------------------------
> > > > > > > Mobj correct       ||  A. 15005  (95%)   |         C. 777 (5%)          |       15782 (100%)
> > > > > > > Mobj incorrect    ||  B.  7764  (58%)    |         D. 5513 (42%)     |       13277 (100%)
> > > > > > >
> > > > > > > When Mobj's output is equal to the ground-truth output (the first row), it is almost always used. When its output is not correct, it is less likely to be used. Note that B. may seem high because the questions studied (questions starting with 'what') likely need some information from objects, and also Mobj is doing a 1600-way classification, therefore it is not always easy for Mobj to be correct (e.g. if the ground truth answer is 'monitor' and Mobj outputs 'tv', it would be considered as incorrect).
> > > > > > >
> > > > > > >
> > > > > > > We thank the reviewer for all the interesting suggestions. It seems that the concerns have been addressed, and we hope it will reflect in the reviewer's final rating.

---

### Official Review · AnonReviewer2 · 2018-11-03
**The paper proposes to combine different task-level modules in a progressive way for the task for VQA. The model achieved state-of-the-art performance.**

**Rating:** 7
**Confidence:** 5

**Review:**

The paper proposes to learn task-level modules progressively to perform the task of VQA. Such task-level modules include object/attribute prediction, image captioning, relationship detection, object counting, and finally VQA model. The benefit of using modules for reasoning allows one to visualize the reasoning process more easily to understand the model better. The results are mainly shown on VQA 2.0 set, with a good amount of analysis.

- I think overall this is a good paper, with clear organization, detailed description of the approach, solid analysis of the approach and cool visualization. I especially appreciate that analysis is done taking into consideration of extra computation cost of the large model; the extra data used for visual relationship detection. I do not have major comments about the paper itself, although I did not check the technical details super carefully.

- One thing I am confused about is the residual model, which seems quite important for the pipeline but I cannot find details describing it and much analysis on this component.

- I am in general curious to see if it will be beneficial to fine-tune the modules themselves can further improve performance. It maybe hard to do it entirely end-to-end, but maybe it is fine to fine-tune just a few top layers (like what Jiang et al did)?

- One great benefit of having a module-based model is feed in the *ground truth* output for some of the modules. For example, what benefit we can get if we have perfect object detection? Where can we get if we have perfect relationships? This can help us not only better understand the models, but also the dataset (VQA) and the task in general.

---

> ### Author Response · Authors · 2018-11-11
> **Response to Reviewer 2**
>
> We thank the reviewer for the comments and feedback. We will also include the suggested experiment that shows the plug-and-play nature of PMN.
>
> 1. Residual modules
> - Residual modules are small neural networks (e.g., an MLP for Mvqa, Sec. 3.4, (4)) that a task module may use when other lower level modules are incapable of providing a solution to a given query. For example, consider the question “is this person going to be happy?” on an image of a person opening a present. Lower level modules of Mvqa may not be sufficient to solve the question. Therefore, Mvqa would make use of its residual module, which would essentially learn to “pick up” all queries that lower level modules cannot answer.
>
> 2. Effect of fine-tuning
> - While it might be beneficial to fine-tune the modules for a specific parent task we want each module to be an expert for their own task as it facilitates a plug-and-play architecture. Fine-tuning may push the modules towards blindly improving parent module’s performance but (i) badly affect interpretability of inputs and outputs; and (ii) may also reduce the lower module’s performance on its own task. Most importantly, it would not scale with the number of tasks, as for each task the agent would need to keep several fine-tuned modules of the lower tasks in memory.
>
> 3. Feeding in the ground-truth
> - Thanks for this great suggestion. We performed an experiment where we evaluate the benefits that the VQA model may achieve by using ground-truth captions instead of captions generated by the caption module. Our preliminary experiments show a gain of about 2.0% which is a relatively high gain for VQA.
> This points to important properties of the PMN allowing human-in-the-loop type of continual learning, where a human teacher can pinpoint flaws in the reasoning process and potentially help the model to fix them.

---

> > ### Comment · AnonReviewer2 · 2018-12-12
> > **Thanks for the response**
> >
> > Thanks for the response! It is interesting that the GT captions can help improve the VQA performance, please incorporate the results and update the manuscripts accordingly.
> >
> > Again, I think this is a good paper and will not change my rating.

---

> > > ### Author Response · Authors · 2018-12-12
> > > **Thank you.**
> > >
> > > We have added the GT captions experiment in the 'plug-and-play architecture' paragraph in Section 4.1.
> > >
> > > Thank you again for your great suggestion!

---

### Official Review · AnonReviewer3 · 2018-11-04
**very interesting work, but a lot of the details are not clear.**

**Rating:** 6
**Confidence:** 4

**Review:**

[Summary]
This paper presents a multi-task learning approach for VQA that represent a solver for each task as a neural module that calls existing modules in a program manner. The authors manually design the task hierarchy and propose a progressive module network to recursive calls the lower modules and gather the information by soft-attention. The final prediction uses all the states and question to infer the final answers. The authors verify the effectiveness of the proposed method on the performance of different tasks and modules. Experiment on VQA shows the proposed model benefits from utilizing different modules. The authors also qualitatively show the model's reasoning process and human study on judging answering quality.

[Strength]
1. The proposed method is novel and explores to use the existing modules as a black box for visual question answering.  This is different from most existing work.

2: By examing different modules, the proposed method is more interpretable compare to canonical methods.

3: The experiment results are good, especially for the counting problem.

[Weakness]
1. The title of the paper is "visual reasoning by progressive module networks." The title may be a little overstated since the major task is focused on visual question answering (VQA).

2. Annotation is not clear in this paper. For example, on page 3, Query transmitter and receiver, "the output o_k = M_k(q_k) received from M_k is modified using receiver function as v_k = R_{k->n}(s^t, o_k). " There are multiple new variables in this paragraph, without specifying the dimension and meaning for each attribute, it's really hard to understand. On page 4, State update function, what is the meaning of variable "Epsilon" in the equation? From the supplementary, it seems Epsilon means the environment?

3. On the object counting task, the query transmitter needs to produce a query for a relationship module. The authors mentioned that this is softly calculated by softmax on the importance score. Since q_rel require one hot vector as input, how to sample the q_rel given the importance score and how backprob the gradient in this case?

4. The cider score of image captioning is 109 compared to the baseline 108. The explanation is the COCO dataset has a fixed set of 80 object categories and does not benefit from training the diverse data. Since the input visual feature is the same, the only difference is the proposed model has additional label embedding as input. My assumption is the visual feature already contains the label information for image captioning.

5. On relational detection task, is there a way to compare with the STOA method on some specific data split? This will leads to much more convincing results.

6. Similar as above question, on the object counting task, is there a way to compare with previous counting methods?

7. In Table 4, the accuracy of number on Zhang et.al is 49.39, which is higher than other methods, while on test-dev, the accuracy is 51.62, which is lower than others. Is the number right?

---

> ### Author Response · Authors · 2018-11-11
> **Response to Reviewer 3**
>
> We thank the reviewer for the comments and feedback. We will certainly clarify them in the final paper.
>
> 1. Title of the paper
> - We agree that the main highest-level task that we show is VQA, even though our method is more general. Our title aimed to convey that we showcase PMN on a host of increasingly complex visual reasoning tasks such as relationship detection, counting, and captioning, as well as VQA. Our focus is on VQA as it happens to be one of the most complex visual reasoning tasks that can leverage each of the (relatively) simpler tasks.
>
> 2. Description of variables
> - Thanks for the feedback. Epsilon means the environment, some of the definitions are written in Section 3, but we agree that it can be somewhat challenging to interpret as there are many variables. We edited the text to address variables more gently and to explain the arrow sign.
>
> 3. Query for the relationship module
> - The relationship module is fed an N-dimensional (corresponding to N image regions) one-hot vector as input during training. When it is called by other task modules (such as counting), an N-dimensional probability vector is computed using softmax on image regions (see A.4, point 3) and not using the importance scores. This acts as a soft version of the one-hot sampled vector so that we can backpropagate gradients.
>
> 4. CIDEr score of captioning
> - That may be true to some extent. However, we think that explicit label information might still be useful since the visual features (environment) are from Faster RCNN and contain diverse information such as edges, background, color, and size.
>
> 5 and 6. Comparison with SOTA models for counting and relationship detection
> - To the best of our knowledge, Zhang et al. (2018) is the SOTA method on counting in the context of visual question answering. Our counting module leverages that but achieves higher performance on the number questions - 54.39% with ensembling and 52.12% without vs. 51.62% of Zhang et al. (2018). Note that 51.62% of Zhang et al. (2018) is from a single highly regularized model that provides small gains from ensembling. This shows that additional modules help. Kim et al. (2018) which is concurrent to our work shows similar performance. For the relationship detection task, other works such as Lu et al. (2016) unfortunately have a different setup which makes direct comparison difficult.
>
> 7. Table 4, accuracies are from Zhang et al. 2018
> - Yes, the numbers are from their paper. One possible explanation for this could be their use of high regularization for a single model instead of ensembling. Thus, the performance improvement from training on the train set (evaluating on validation) to training on train+val (evaluating on test-dev) is smaller.
>
> (Zhang et al. 2018) Learning to Count Objects in Natural Images for Visual Question Answering
> (Kim et al. 2018) Bilinear Attention Networks
> (Lu et al. 2016) Visual Relationship Detection with Language Priors

---

### Author Response · Authors · 2018-11-23
**Revision**


1. We clarified some notational ambiguities pointed out by Reviewer 3.

2. We added an experiment demonstrating the plug-and-play nature of PMN as suggested by Reviewer 2.

3. As we do not claim PMN is a generalization of Neural Module Networks, we edited the paper to remove misunderstanding our wording may have caused.

We thank all reviewers for their valuable feedbacks.

---

### Meta-Review · Area_Chair1 · 2018-12-14

**Confidence:** 5
**Recommendation:** Accept (Poster)

**Metareview:**

Important problem (modular & interpretable approaches for VQA and visual reasoning); well-written manuscript, sensible approach. Paper was reviewed by three experts. Initially there were some concerns but after the author response and reviewer discussion, all three unanimously recommend acceptance.